# Rank-Then-Act: Reward-Free Control from Frame-Order Progress

## Abstract

We introduce Rank-Then-Act (RTA), a novel reward-free control framework that enables policy learning without extrinsic task rewards. Instead, RTA uses a progress-percentage signal derived from expert video demonstrations (evaluated via rank correlation). Specifically, we train a Vision–Language Model (VLM) progress scorer offline with a Group Relative Policy Optimization (GRPO) objective, assigning progress percentages to shuffled frames from expert gameplay. This scorer is then frozen and used to provide feedback during reinforcement learning (RL). During policy learning, the agent receives as reward the Spearman correlation coefficient between the VLM scorer's predicted progress percentages for a window of recent observations and their true environment timestamps, yielding a bounded, time-aligned progress signal without explicit task rewards. On the PyBoy Catrap and Kirby environments, RTA enables a VLM-based agent to solve multiple levels of varying difficulty using only expert videos, without reward engineering. We compare against Rank2Reward and VLM-RM baselines trained on the same video-only data, and show that a single progress scorer trained once can be reused across levels and games without per-task video tuning.. Our results demonstrate that training VLMs to act in games without extrinsic rewards is a promising and scalable direction for generalizing RL to settings where reward specification is impractical or impossible.

## 1 Introduction

Learning control from pixels *without* extrinsic rewards is a foundational challenge for generalist agents, particularly in domains where reward design is brittle, unavailable, or easily exploited-such as retro games, simulators with weak task APIs, and real-world robotics, where proxy rewards often lead to specification gaming. Training multimodal agents based on vision–language models (VLMs) offers a promising path: rather than relying on hand-crafted rewards, we can derive a dense notion of *progress* directly from expert video snippets and optimize VLM policies with respect to this signal. However, three persistent obstacles remain: (i) chronological inputs invite trivial shortcuts (*later is better*), yielding vacuous, monotonically increasing progress scores; (ii) absolute progress scales are ambiguous across tasks and episodes; and (iii) online learning requires reward signals that are inexpensive to compute, sufficiently informative to guide exploration, and robust to distribution shift. In this paper, we develop an approach to overcome these obstacles.

We introduce **Rank–Then–Act** (RTA), a two-stage framework that transforms *frame-order* information from expert video into a robust **progress-percentage** signal (evaluated via progress–time correlation) for online control. RTA is designed to train multimodal agents that learn complex behaviors directly from environments without extrinsic task rewards. *Stage 1* trains-once and offline-a vision–language **progress scorer** (Hu et al., 2022; Bai et al., 2025) with a **Group Relative Policy Optimization (GRPO)** objective (Shao et al., 2024b), enabling the model to **assign progress percentages** to frames within *shuffled* short expert video clips. Shuffling explicitly breaks temporal shortcuts, forcing the **scorer** to rely on genuine, task-relevant visual cues rather than absolute time (Misra et al., 2016; Ma et al., 2025). Additionally, anchor conditioning (see §3.2) provides a shared temporal reference while ensuring permutation invariance across **scored** frames. After training, the **scorer** is frozen. *Stage 2* deploys this pretrained estimator online: at each time step, we form a window of recent observations, use the VLM **scorer to predict progress percentages**, and compute the **progress–time correlation**-Spearman's $\rho$ between the predicted **progress percent-**

**ages** and the true environment timestamps. This scalar correlation serves as the *sole* reward signal for Proximal Policy Optimization (PPO) (Schulman et al., 2017) adaptations to VLMs; no environment rewards are used at any point. This design enables policy learning driven entirely by progress inferred from expert demonstrations and environment interactions.

Existing approaches only partially address this setting. Video-based reward models and VLM reward heads typically require engineered goal text, per-task tuning, or produce uncalibrated scalar values that are hard to reuse. Reward-free RL and intrinsic motivation methods assume reward queries at deployment and do not capture progress from demonstrations. Most imitation-from-observation and video IRL methods require action labels or environment rewards, which are not assumed in our setting. RTA fills this gap by providing a simple, scale-free progress signal from expert videos, using only rank correlation with time as a reward for policy learning.

Our main contributions are as follows:

1. We introduce a simple yet effective Vision–Language Model (VLM) **progress scorer**, trained offline on shuffled expert video clips with a GRPO objective, taught to assign **progress percentages** on shuffled clips, and kept frozen during policy learning.

2. We propose a correlation-only reward signal for online control, computed as Spearman correlation between predicted **progress percentages** and time indices over sliding windows-enabling dense, shaped, reward-free feedback.

3. We empirically demonstrate, on multiple PyBoy games (Catrap and Kirby) and levels, that this rank-correlation signal enables agents to solve tasks without any environment rewards, outperforming supervised fine-tuning and other methods, both when using a progress scorer trained per level and when reusing a single progress scorer across tasks without per-level or per-game video tuning..

RTA contributes a simple, correlation-only reward design that enables VLM-based policies to learn in fully reward-free environments from video alone, and we provide direct comparisons to prior VLM reward models and ranking-based video rewards.

## 2 RELATED WORK

**Ordinal progress and ordering-based supervision from video.** A growing line of work uses *ordering* as supervision to obtain shaped rewards from passive videos. Rank2Reward Yang et al. (2024) similarly learns shaped rewards from passive video via temporal ranking. In our experiments, we adapt Rank2Reward as a non-VLM ranking baseline using the same game video data and compare its learned rewards and downstream control performance to RTA's correlation-only signal. VLM-RM Rocamonde et al. (2024b) uses a clip-based progress scorer by comparing the goal completion frame with the current frame. In the original paper, it was shown that this approach often fails out-of-distribution and requires per-task environment tuning, such as changing textures or adjusting the view angle. Related efforts learn progress-sensitive embeddings or sequence rewards via temporal constraints and alignment (Zakka et al., 2021a; Wang et al., 2024). *RTA* is closest in spirit to Yang et al. (2024) but differs in that (i) we fine-tune a compact LoRA *Vision–Language Model (VLM)* once with a *listwise* objective on *shuffled* clips (with *anchor conditioning*) to output per-frame *progress percentages*, and then *freeze* it; and (ii) we reward via *Spearman progress–time correlation* (correlation between predicted progress percentages and timestamps) over a *sliding window*, rather than learning a scalar progress head or using adversarial training.

**Vision–language models as reward/value estimators.** Pretrained VLMs have been used as zero-shot reward models by scoring observations against natural-language goals or ordering frames to estimate value (Rocamonde et al., 2024a), and as universal progress/value estimators by reframing value prediction as frame ordering over shuffled clips (Ma et al., 2025). VLAC Zhai et al. (2025) train specific critic by using the same logic provided by GVL. These approaches leverage world knowledge in VLMs to avoid task-specific reward engineering. *RTA* similarly capitalizes on a VLM prior but (i) fine-tunes it *offline* as an ordinal *progress scorer* on expert clips (outputting progress percentages) and (ii) converts its outputs into a correlation-based, sliding-window reward, enabling stable control from video alone without extrinsic environment rewards. We train the scorer with a *GRPO-style listwise preference objective* on *shuffled* clips; shuffling blocks "later-is-better" shortcuts and forces reliance on task-relevant visual cues.

**Reward-free RL and intrinsic motivation.** Classical reward-free RL formalizes a two-phase protocol: first explore without rewards, then solve downstream reward queries using the collected data (Jin et al., 2020; Kaufmann et al., 2021; Chen et al., 2022). In parallel, intrinsic-motivation methods (e.g., curiosity or prediction-error bonuses) provide task-agnostic signals that enable progress in sparse- or no-reward regimes (Pathak et al., 2017). These approaches emphasize state-space coverage or generic exploration rather than learning a task-specific progress signal from demonstrations. Our setting differs: we avoid extrinsic rewards and do not answer reward queries; instead, we optimize order consistency from demonstrations. *RTA* derives a demonstration-driven *ordinal* progress signal from expert video and optimizes *only* a progress–time correlation objective, without coverage objectives or reward queries.

**Imitation from observation and IRL from video.** When actions or rewards are unavailable, imitation-from-observation and IRL learn behaviors from state/video-only data. Adversarial formulations such as *Generative Adversarial Imitation Learning (GAIL)* and *Generative Adversarial Imitation from Observation (GAIfO)* learn policies by distribution matching without explicit reward engineering (Ho & Ermon, 2016; Torabi et al., 2019). Beyond frame-wise matching, video-based methods handle temporal misalignment and embodiment gaps via sequence-level objectives and cross-video temporal constraints, including XIRL (Zakka et al., 2021a), ordered-coverage alignment (ORCA) (Huey et al., 2025), alignments based on *Soft Dynamic Time Warping (Soft-DTW)* (Wang et al., 2024), and time alignment via video matching (Haresh et al., 2021). *RTA* departs from adversarial or regression-style reward learning by (i) training a listwise VLM progress scorer offline, then freezing it; and (ii) rewarding online control by the *Spearman* correlation between the scorer's progress percentages over a recent window and true timestamps-avoiding *both* adversarial training *and* scalar reward regression. Methods such as T-REX/D-REX Brown et al. (2019b;a) and XIRL/ORCA Mukherjee et al. (2023); Zakka et al. (2021b) operate in a different data regime: they require multiple demonstration trajectories of the same task or decompose the task into subgoals, whereas our setting assumes only raw videos with no actions or rewards. Adapting them would require either recovering actions from human play or changing the problem formulation. We therefore focus our empirical comparison on video-only reward models and ranking baselines (Rank2Reward, VLM-RM) that can be applied without action labels, and leave adapting action-dependent IRL methods to video-only VLMs for future work. VICtoR Hung et al. (2025) is also not directly comparable to our setting, as it is a vision–instruction correlation method that relies on natural-language task descriptions and additional supervision (e.g., motion-level annotations and object/state signals).

## 3 METHOD

We present Rank-Then-Act (RTA), a two-stage framework that converts a model's grasp of frame order in expert videos into a bounded learning signal in $[-1, 1]$ for reward-free control. Stage 1 (Rank): fine-tune a VLM as a listwise progress scorer that assigns each frame a progress percentage (larger = later), trained with GRPO (Shao et al., 2024b) using Spearman's rank correlation between predicted scores and ground-truth timestamps as the reward. Stage 2 (Act): freeze the scorer and, during interaction, at query steps compute Spearman's $\rho$ over a sliding window between the scorer's outputs and the window's timestamps; this correlation is the sole reward for policy-gradient training of the VLM-based agent (no extrinsic environment rewards). We next formalize the control setting and define this correlation-based reward primitive.

### 3.1 PROBLEM SETTING AND CORRELATION PRIMITIVE

We consider goal-conditioned, partially observed Markov decision processes with observation space $\mathcal{S}$, action space $\mathcal{A}$, transition function $\mathcal{P}$, episode horizon $\mathcal{T}$, and a textual goal $g$. The goal $g$ is supplied as part of the agent's textual prompt to initialize the task. We do not, however, condition on $g$ in the conventional RL sense; $g$ is simply included in the prompt for goal-directed tasks. The agent $\pi_\theta : \mathcal{S} \to \Delta(\mathcal{A})$ maximizes expected return using an implicit, model-derived reward. When extrinsic rewards are absent, learning relies solely on an estimated measure of *progress consistency* produced by a model trained on expert video demonstrations.

Our sole scalar signal in both stages is the Spearman rank correlation,

$$\mathrm{spr}(x, y) := \mathrm{Pearson}(\mathrm{rank}(x), \mathrm{rank}(y)) \in [-1, 1],$$

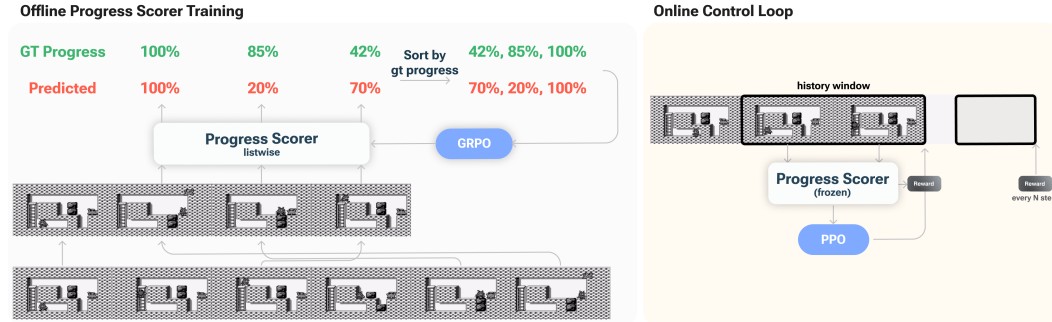

Figure 1: **RTA: reward-free control in two stages.** *Stage 1* trains a VLM *progress scorer* on shuffled clips with a GRPO objective that maximizes *progress–time* Spearman $\rho$. *Stage 2* uses the frozen scorer online: every $N$ steps, we score a window (with $L$ shuffles) and use the average $\rho \in [-1, 1]$ as the *only* reward for PPO-style learning.

We use $\mathrm{spr}(\cdot, \cdot)$ both as the bounded, task-agnostic training objective in Stage 1 and as the reward signal in Stage 2. The idea of using rank correlation follows Ma et al. (2025) as an estimator of trajectory correctness.

## 3.2 STAGE 1: LISTWISE PROGRESS SCORER VIA GRPO

**Expert clip batching.** Given an expert video demonstration $\tau = (s_1, \ldots, s_T)$, we segment it into frames $(s_1, \ldots, s_{\tilde{K}})$, which we then use to construct training instances for progress prediction. Following Ma et al. (2025), to avoid the shortcut "later is always better," we *anchor* the first frame and randomly shuffle the remaining frames:

$$(s_{\mathrm{anc}}, s_{\tilde{2}}, \ldots, s_{\tilde{K}}), \qquad (\tilde{2}, \ldots, \tilde{K}) = \texttt{permute}(2{:}K), \quad s_{\mathrm{anc}} := s_1.$$

Only the shuffled, non-anchor frames are scored; the anchor provides a fixed starting point for each training sequence. Fully shuffling frames can make the direction of progress ambiguous or overemphasize local appearance changes. Anchoring breaks this symmetry and encourages the model to evaluate true *progress* with respect to the task, rather than merely tracking local appearance drift. The same anchoring and shuffling procedure is applied at inference time.

**VLM input and output.** A vision–language model $f_\phi$ receives a sequence of $K$ frames as input and is prompted to generate, for each (shuffled) frame excluding the anchor, a reasoning trace that specifies its predicted *progress percentage* within the original sequence (higher = later). The canonical output format is:

```
Frame i:  ... Frame Description:  ... Progress:  p_i%.
```

Collecting these $K-1$ scalars yields the vector $\mathbf{p} = (p_2, \ldots, p_K)$, where each $p_i$ denotes the predicted progress percentage for frame $i$ (anchor fixed at position 1 and never scored). All outputs are parsed using a strict regular expression. If parsing fails-for example, when the number of parsed scores does not equal $K-1$-we assign the minimum possible reward, $R_{\min} = -1$. For non-numeric or unparsable outputs, we treat parsing as failed.

**Listwise reward and target.** Let $\mathbf{p}$ denote the predicted *progress percentages* for the shown (shuffled) frames, and let $\mathbf{q}$ be their ground-truth temporal indices from original timestamps. We define the reward as

$$R = \mathrm{spr}(\mathbf{p}, \mathbf{q}), \tag{1}$$

where $\mathrm{spr}(\cdot, \cdot)$ is Spearman's rank correlation. Using Spearman on continuous progress percentages makes the objective insensitive to the specific scale of $p_i$ while rewarding correct monotone order.

**GRPO objective and optimization.** We treat the VLM's text generation process as a sequential policy over output tokens, and optimize a Group Relative Policy Optimization (GRPO) objective (Shao

et al., 2024b). The GRPO loss is:

$$\frac{1}{|a|} \sum_{i=1}^{|a|} \min\Big( r_i \hat{A}, \ \mathrm{clip}(r_i, 1 - \epsilon, 1 + \epsilon) \hat{A} \Big)$$

where $a$ denotes the generated action (token) sequence, $r_i = \pi_\theta(a^i \mid s, a^{<i})/\pi_{\theta_{\mathrm{old}}}(a^i \mid s, a^{<i})$ is the importance weight, $\epsilon$ is a clipping parameter, and $\hat{A}$ is the relative group advantage. This advantage $\hat{A}$ is defined as in Shao et al. (2024b), based on the maximized expected reward:

$$\max_\phi \ \mathbb{E}_{(\mathrm{segment, \ shuffle})} \ \mathbb{E}_{\mathrm{output} \sim f_\phi} \big[ R \big]$$

where $R$ is the progress–time Spearman correlation reward defined above. After convergence, the progress scorer $f_\phi$ is frozen for use in Stage 2.

### 3.3 STAGE 2: ONLINE CONTROL FROM PROGRESS–TIME CONSISTENCY

**Inference window, anchor, and reward.** At each environment step $t$, construct the window $\mathcal{W}_t = (s_{t-m+1}, \ldots, s_t)$ with $m = \min(N, t)$, and designate the oldest state $s_{t-m+1}$ as the *reference anchor* $s_{\mathrm{anc}}$. We query the frozen scorer only on *query steps* ($t \bmod N = 0$ or $t = \mathcal{T}$); otherwise we set $r_t = 0$. On each query step, to reduce VLM cost and variance, draw $L$ independent permutations of the $m - 1$ non-anchor frames (keeping $s_{\mathrm{anc}}$ fixed), obtain progress percentages from the scorer for each permuted window, compute their progress–time correlations, and average these correlations to yield the scalar reward $r_t$. Unless stated otherwise, we use $L = 4$ and $N = 15$.

**Policy optimization and normalization.** We train the policy $\pi_\theta$ using Proximal Policy Optimization (PPO) variants tailored for VLMs and LLMs, using $r_t$ as the sole reward signal. The reward is bounded in $[-1, 1]$ by construction. Since rewards are sparse (computed every $N$ steps) and bounded, we follow VL-DAC(Bredis et al., 2025) use Generalized Advantage Estimation (GAE) based on the observed $r_t$ sequence.

For experiments with LOOP (Chen et al., 2025)-an extension of GRPO to multi-turn settings-we set the episode length to $\mathcal{T} = N$ and compute the reward only at the end of each episode, assigning the same advantage to all output tokens. Because this limits long-horizon environment interaction, we restart the trajectory from a new starting point whenever the average episode reward $r_\mathcal{T}$ across sampled traces exceeds a threshold $\tau$. At each reset, we select the trace with the highest reward and repeat the process.

## 4 EXPERIMENTS

We evaluate RTA as a reward-free approach to vision–language policy learning in game-like environments. Agents receive only expert video demonstrations; no extrinsic rewards, environment APIs, or task annotations are used. The evaluation mirrors our two-stage method: (i) train and analyze a listwise VLM progress scorer that assigns each frame a progress percentage; (ii) optimize a control policy using only the scorer's progress–time correlation as reward. This setup tests two hypotheses: VLMs can reliably infer ordinal progress from video, and these signals alone suffice to train agents to solve complex tasks.

### 4.1 SETUP

**Expert videos.** For Stage 1, we use human *Catrap* playthroughs (levels 1–8; successful episodes), one *full-game* Catrap run, and 70 additional GameBoy playthroughs scraped from YouTube. Videos are split into fixed-length clips of $K$ consecutive frames and converted to the *anchor+shuffle* interface (Sec. 3.2). To standardize inputs and avoid cues such as completion time or lives, Catrap frames are cropped from $640 \times 360$ to $640 \times 330$. We also run ablations on other games. Figure 2 reports validation *progress–time* Spearman $\rho$ when training on Catrap levels (1–8) and on full playthroughs of Catrap, Donkey Kong, and Kirby. For level curves we sample ordered frames from non-overlapping within-level clips; for full runs we sample uniformly at 0.5 FPS. Training is stable across settings; later levels converge more slowly and to slightly lower asymptotes, likely due to higher visual/temporal complexity.

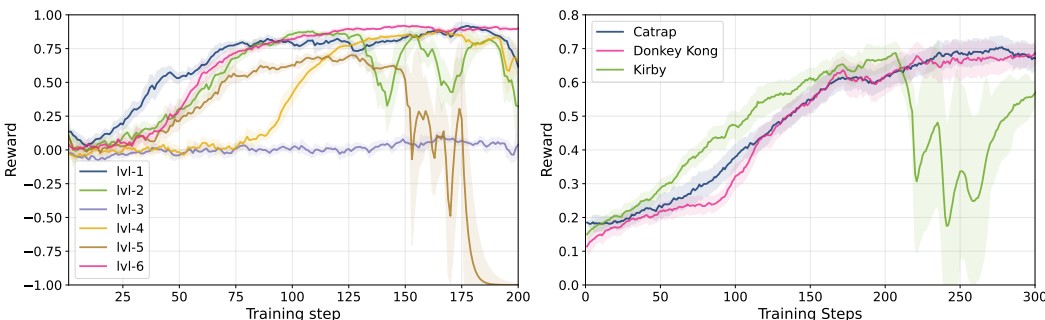

Figure 2: **Stage 1 learns reliable ordinal progress. Left:** per-level Catrap training shows fast rises in *progress–time* $\rho$ with slower convergence on intricate levels. **Right:** training on full playthroughs across games also converges, with slightly lower plateaus due to non-informative segments (e.g., menus).

**Tasks and levels**. For control evaluation, we consider multiple Catrap levels (L2, L4, L6, with progressive difficulty) and a second game, Kirby (part until the first boss is met, corresponding to the first checkpoint in VideoGameBench (Zhang et al., 2025), we'll call it level-0 for simplicity), in the PyBoy emulator. Each task is defined by a goal description and termination condition, and we train separate policies per task while reusing a trained (from per-task to general) Stage-1 progress scorer. This setup stresses both cross-level and cross-game reuse of the scorer, and enables us to probe robustness of the progress–time correlation reward beyond a single level.

**Observations and actions.** For Stage 2 policy learning, we render environment observations at $160 \times 144$ pixels; for VLM policy inputs, frames are centrally cropped and resized to $160 \times 120$. The action space consists of the four D-pad directions (up, down, left, right) defined by PyBoy's discrete control interface. To ensure consistency with Stage 1, we resize frames back to the original training resolution before querying the frozen scorer $f_\phi$ to obtain progress percentages, which we then correlate (*progress–time* Spearman) with timestamps. For Kirby we allow pressing multiple buttons simultaneously, following the VideoGameBench Light Zhang et al. (2025) setup.

**Technical details.** In both stages, we train only the LoRA adapters (rank 16, alpha 32) on top of the provided models. For Stage 1 scorer training, each reported experiment requires 4 H100 GPUs, and 200 training steps take approximately X hours. For Stage 2 policy learning (using either VL-DAC or LOOP), 1 H100 GPU is required, and training takes up to 24 hours. Detailed hyperparameter settings are provided in Appendix A.

### 4.2 STAGE 1: PROGRESS-SCORER LEARNING DYNAMICS AND GENERALIZATION

We begin by verifying that a single VLM-based scorer can learn to score progress within shuffled expert clips across different levels, and we investigate its ability to generalize across levels, games, and domains.

**Per-level training curves.** We first train the scorer on individual Catrap levels. The left panel of Fig. 2 shows the GRPO training curves for the first six levels. Most levels exhibit rapid convergence within 200 steps, achieving asymptotic *progress–time* Spearman $\rho$ near $0.9$. Some training runs diverge after reaching high performance-a phenomenon also observed in extended GRPO training, and consistent with known GRPO instabilities. All training videos in this experiment were extracted under human supervision.

As shown in A.1, level 3 displays alternating periods of convergence and divergence, suggesting that even in levels with less clear task goals, progress can be learned, albeit with greater sensitivity to data preparation and training duration.

**Full-playthrough training.** Next, we pool all Catrap levels (1–8) and train a *single* scorer across this part of game. The right panel of Fig. 2 shows training $\rho$ on a mixed-level and games held-out

set. In these experiments, we train the model for 300 steps on three different games to achieve and sustain high asymptotic quality.

Pooling all levels increases data diversity and produces smoother training dynamics, though the final performance plateau is slightly lower than in most single-level runs. This drop is likely due to segments with static or non-informative frames (such as menus) present in full playthroughs. Importantly, this experiment demonstrates that a single progress scorer can be trained end-to-end from full-game videos, obviating the need for level-specific tuning and opening a path toward scalable, domain-general learning. Additional results confirm that per-game training also provides robust performance on level-specific validation.

**Cross-level generalization.** We train the progress scorer on a *source* set-either a single level $i$, a full-game playthrough, or a pool of 70 scraped GameBoy playthroughs-and evaluate on a disjoint *target* level $j$. Performance is the Spearman correlation between predicted progress and time (*progress–time* $\rho$). Figure 3 shows the mean validation $\rho$ for all source–target pairs $(i \rightarrow j)$: diagonal cells (within-level) give upper bounds, while off-diagonals measure transfer.

Transfer is notably asymmetric: scorers trained on any level generalize especially well to level 1, likely due to its lower complexity. The strongest overall transfer comes from pooled training on all levels, indicating that a single, shared scorer can support domain-general inference-from full-game playthroughs to individual levels. Training on a diverse mix of 70 GameBoy playthroughs also yields solid zero-shot performance on held-out levels, despite limited direct exposure.

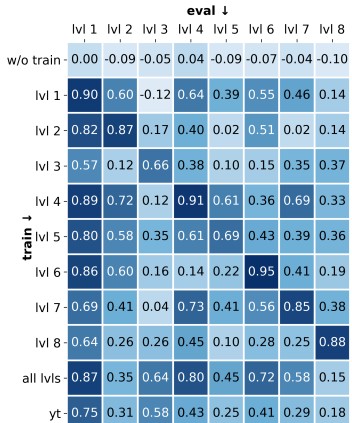

Figure 3: **Cross-level transfer of ordinal progress.** Heatmap of validation *progress–time* $\rho$ for source (rows) → target (columns). Diagonals are in-level; off-diagonals show transfer. Pooled training (all levels or diverse games) transfers best; simpler Level 1 generalizes from most sources.

Both per-level and pooled setups achieve high in-distribution $\rho$. We use separate per-level scorers, per-game scorers, and a scorer trained on different YouTube playthroughs for our main Stage 2 Catrap and Kirby control experiments. We also conduct Stage 2 experiments with the Stage 1 scorer trained in different data setups (per-level, per-game, and general). In these experiments, we use a single scorer trained on pooled Catrap and Kirby playthroughs and YouTube data; in this setup, we do not perform any per-level or per-game fine-tuning of the progress model, and this is sufficient for the Stage 1 ranker to produce a good rank-based reward. For more detailed results, refer to the **cross-domain generalization** section.

### 4.3 STAGE 2: CONTROL FROM PROGRESS–TIME CONSISTENCY

We now deploy the frozen progress scorer in the environment and train a policy using policy-gradient methods, optimizing *only* the progress–time correlation reward $r_t$ (see Sec. 3.3), with no access to any extrinsic task rewards.

**Training dynamics (reward).** Figure 4 shows the evolution of the per-query *progress–time* Spearman reward during training on Catrap Level 2. Although the reward is computed sparsely (every $N = 15$ steps) and is always bounded in $[-1, 1]$, it provides sufficiently shaped feedback for policy learning: the mean reward increases steadily throughout training, closely tracking the agent's success rate. In the plot, we show results from two random seeds, with quality smoothed with an EMA over the last 10 *queries*.

This experiment demonstrates that the progress–time correlation signal-derived solely from video-based ordinal progress-is both informative and stable enough to drive effective online RL, even in the complete absence of extrinsic rewards. We also conduct experiments on Catrap level 4, level 6, and the Kirby game. Levels 4 and 6 require advanced reasoning and backtracking from the model, as solving these levels involves overcoming dead ends. Kirby is a long-horizon control game. The corresponding results are presented in Table 1.

**Task success.** Figure 4 also reports the *success rate* (rolling window over episodes) alongside the per-query *progress–time* reward when optimizing with VL-DAC (Bredis et al., 2025). As highlighted

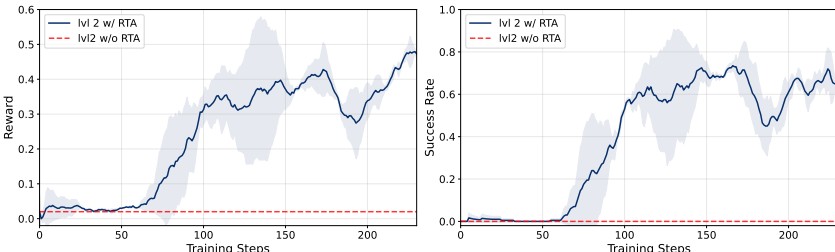

Figure 4: **Correlation-only reward drives control.** On Catrap L2, L4, L6, the per-query *progress–time* $\rho$ (computed every $N = 15$ with $L = 4$ shuffles) rises during training, and episode success increases in step, despite bounded, sparse rewards and no extrinsic signals.

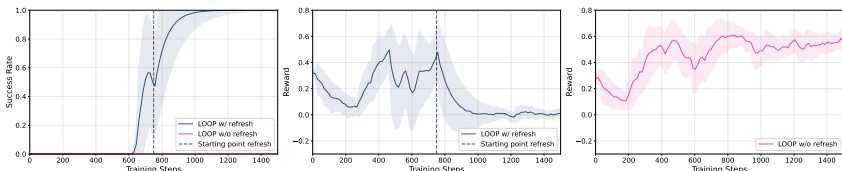

Figure 5: **LOOP benefits from starting-point refreshing.** Without refreshing, rewards improve but may not reach success when $N$ is shorter than the task. Refreshing from the best terminal state after threshold $\tau$ enables completion; brief post-refresh dips arise if $N$ exceeds remaining steps.

in the abstract and Methods, RTA achieves a high success rate on Catrap Level 2, 4, 6, while the baseline model, lacking any reward signal-remains at $0\%$, confirming that a purely *progress–time (ordinal)* signal is sufficient to drive effective policy learning in the absence of any environment reward. At several points during training, the RTA-trained agent attains $100\%$ success (if we turn off smoothing). Additionally, we run Stage 2 experiments on the Kirby game, where RTA also achieves a positive success rate. In contrast, most modern frontier VLMs are unable to complete this game (Zhang et al., 2025), so even modest success distinguishes our approach from these models., and even tiny success become different to frontier models.

We plot both the progress–time reward and the success rate for direct comparison; both metrics increase steadily and in pair, further validating the coherence and shaping power of our reward signal. Additionally, we plot the correlation between the success rate and the reward obtained from Stage 1 to demonstrate the strong relationship between task completion and the provided reward signal.

**LOOP policy learning.** Figure 5 presents the per-query *progress–time* Spearman reward during training with LOOP (Chen et al., 2025). We report two reward curves: one for training with **starting point refreshing** enabled, and another without refreshing.

When the model is trained without refreshing the starting point, rewards increase steadily, but there is no corresponding rise in success rate. This occurs because the default window length $N$ is insufficient for the agent to complete the full task within a single episode. To address this, we introduce **starting point refreshing**: after achieving a threshold $\tau$, we select the trajectory with the highest reward and begin the next rollout from the final state of that trajectory. This approach enables the algorithm to achieve high success rates, as reflected by the success curve.

However, if the new rollout's length $N$ exceeds the number of steps required to finish the level, this can occasionally produce a temporary decline in reward after refreshing, as seen in the post-refresh segments of the plot. This effect is due to the agent finishing the level quickly, leaving extra rollout steps with little meaningful scoring signal.

**Comparison with baselines.** In the absence of explicit environment rewards, there is a several approaches for current VLM and LLM-based control is to train on expert video demonstrations using supervised fine-tuning (SFT), Rank2Reward (which has not previously been applied to VLMs) and VLM-RM. A comparison of success rates on Catrap levels 2, 4, and 6, as well as Kirby level 0, can be found in Table 1, where RTA outperforms the other methods. Additionally, to show that RTA

| Task | Level 2 | Level 4 | Level 6 | Kirby (level 0) |
|---|---|---|---|---|
| GVL | 0.47 ± 0.25 | 0.00 ± 0.00 | 0.04 ± 0.08 | 0.00 ± 0.00 |
| VLM-RM | 0.40±0.28 | 0.16 ± 0.20 | 0.08 ± 0.16 | 0.0 ± 0.0 |
| VLM-RM$_{reg}$($\alpha = 0.5$) | 0.44 ± 0.32 | 0.00 ± 0.00 | 0.08 ± 0.16 | 0.0 ± 0.0 |
| Rank2Reward | 0.60 ± 0.28 | 0.20 ± 0.0 | 0.13 ± 0.09 | - |
| Oracle reward | 0.50 ± 0.22 | 0.07 ± 0.09 | 0.2 ± 0.0 | **0.40 ± 0.28** |
| RTA (w stage1 training) | **1.00 ± 0.00** | **0.72 ± 0.35** | **0.32 ± 0.27** | 0.07 ± 0.09 |

Table 1: Comparison of success rates from different approaches. Means and std are computed over 5 seeds. Oracle reward is binary reward according to success of the end of level.

provides a stable reward signal for agent learning, we report the Pearson correlation between mean cumulative reward and pass@5 after each training update for the best-performing run (in terms of success rate) for every method. Results are presented in Table 2.

| Task | Level 2 | Level 4 | Level 6 | Kirby (level 0) |
|---|---|---|---|---|
| GVL | -0.01 | 0.00 | -0.01 | 0.00 |
| Rank2Reward | -0.04 | -0.02 | -0.04 | N/A |
| VLM-RM | 0.53 | -0.19 | -0.33 | 0.00 |
| VLM-RM$_{reg}$ ($\alpha = 0.5$) | 0.25 | 0.00 | -0.16 | 0.00 |
| RTA (w stage1 training) | **0.76** | **0.87** | **0.42** | **0.13** |

Table 2: Pearson correlation of mean cumulative reward and pass@5 during training. We report correlation of 0 if method is unable to solve level and N/A if experiment was not conducted. Results in this table reported across 5 seeds.

For the Catrap environment, we collected expert trajectories across multiple levels and augmented them with artificially generated chain-of-thought (CoT) traces. We then fine-tuned the Instruct version of our model-the same backbone used in RTA, on these CoT-augmented trajectories, and evaluated performance over 200 steps of environment interaction. Full results are shown in Table 4.

| Source | Level 2 | Level 4 | Level 6 | Kirby (level 0) |
|---|---|---|---|---|
| Youtube | 1.00 ± 0.00 | 0.47 ± 0.25 | 0.60 ± 0.28 | 0.20 ± 0.16 |
| Full Catrap | 1.00 ± 0.00 | 0.47 ± 0.38 | 0.60 ± 0.28 | 0.07 ± 0.09 |
| Full Kirby | – | – | – | 0.07 ± 0.09 |

Table 3: Cross-domain evaluation of RTA using different video sources for Stage-1 training. Results in this table report mean ± std across 3 seeds.

**Cross-domain generalization.** To examine the generalization of the Stage 1 scorer across video sources, we carried out additional experiments in which RTA is trained on (a) a single full-game Catrap playthrough, (b) a mixture of YouTube PyBoy playthroughs from multiple games and players (excluding Catrap and Kirby), and (c) a Kirby playthrough. These sources differ in visual appearance (compression artifacts, overlays, distinct play styles, etc.) but share the same underlying game dynamics. While we do not yet evaluate cross-embodiment generalization (e.g., from human egocentric videos to robot states), we do test robustness to moderate visual and stylistic shifts within the same family of games. Results are shown in Table 3.

**Ablations.** We conduct several ablations on window size, reward frequency, and reward averaging. In addition, we experiment with using an MLP backbone for the agent in Stage 2 instead of a VLM, demonstrating that the choice of policy model does not affect training outcomes and that performance gains are primarily due to reward design. Finally, we compare giving rewards after every $N$ steps versus providing a single reward at the end of the trajectory (with samples taken uniformly over the trajectory). Further details are provided in Appendix A.2.

## 5 DISCUSSION

We show that reward-free vision–language policy learning is feasible in discrete-action game environments using only expert video and a rank-based progress signal (Spearman correlation between predicted progress percentages and timestamps). This provides a simple, scalable route to training VLM-driven agents without hand-crafted extrinsic rewards and motivates broader work on multi-modal agency.

By showing that VLMs can learn to act from expert video-without any extrinsic reward engineering-using a progress–time correlation signal, our work points to a scalable paradigm for multimodal agency. Reward-free learning not only broadens the scope of what AI systems can tackle autonomously, it also reduces the risks of reward hacking and unintended behaviors that often arise from poorly specified reward functions. Additionally, we leave the discussion of parsing problems beyond the scope of the current work. For the Stage 1 ranker, which learns ranking in a manner similar to reasoning models, previous work Shao et al. (2024a) has already addressed parsing issues and how leave-one-out policy learning depends on them. For Stage 2, parsing issues have also been discussed in prior work on multi-turn VLM/LLM learning Zhai et al. (2024).

| Task | Level 2 | Level 4 | Level 6 |
|---|---|---|---|
| Base | 0.13 | 0.01 | 0.09 |
| Expert SFT | -0.52 | 0.85 | -0.89 |
| RTA | 0.48 | 0.49 | 0.34 |

Table 4: **Ordinal signal vs. supervised imitation.** Mean *progress–time* Spearman $\rho$ (higher is better) on Catrap L2/4/6 for a base policy (no reward), Expert SFT (from CoT-augmented demos), and RTA (correlation-only reward). RTA outperforms base and achieves strong control without human reward engineering; SFT depends on curated data and varies by level.

We provide examples of both good and bad trajectories, each with their corresponding high or low rewards. We also illustrate how an undertrained Stage 1 scorer can be exploited during Stage 2, showing cases where a good trajectory receives a low reward and a bad trajectory receives a high reward. For further details, see Appendix A.3.

We believe progress in this direction will be crucial for bridging the gap between narrow, task-specific models and robust, generalist agents that can understand and act safely in open, dynamic, and imperfectly specified environments. Since the Stage 1 scorer can rank any number of frames that fit in GPU memory, and we can skip parts of the trajectory via uniform sampling, we can score interaction traces of arbitrary length. Additionally, because we use a VLM as the Stage 1 grader, we rely on its reasoning ability to correctly interpret frame order with respect to the task, which helps relax the monotonic-value assumption and enables application of the Stage 1 scorer to tasks with non-monotonic progress.

**Limitations.** Our evaluation covers only a discrete-action game; we do not demonstrate generalization to multi-level, continuous-control, or highly stochastic settings. The approach depends on sufficient expert video-scarcity or bias can cause level-specific overfitting and hinder transfer (e.g., asymmetric Stage 1 cross-level results). The progress–time correlation signal is local: short windows may reward locally coherent behavior without ensuring global progress or subgoal completion; richer objectives may require goal-conditioned anchors, adaptive windowing, or explicit temporal abstraction. Current VLM backbones add nontrivial latency even with token caching and striding; distilled or more efficient models could help. Despite these limits, the results indicate that ordinal, video-driven rewards are a promising route toward scalable generalist control.

## 6 CONCLUSION

We present Rank-Then-Act (RTA), a simple two-stage method that teaches vision-language models (VLMs) to act using only expert video and progress percentages (via rank correlation), with no extrinsic rewards. In a challenging discrete-action game, this purely reward-free signal trains VLM-based agents that achieve high success rates and outperform strong supervised fine-tuning baselines.

Because real-world rewards are often sparse or hard to design, learning from expert video alone offers a scalable path to multimodal agency while reducing risks like reward hacking. We see this as a step toward robust, generalist agents that can understand and act in open, dynamic, and under-specified environments.

# REPRODUCIBILITY STATEMENT

We release anonymized code, configuration files, and figure notebooks in the supplementary material. The core method is specified in Sec. 3.1 (correlation primitive), Sec. 3.2 (offline progress scorer + GRPO), Sec. 3.3 (online reward/control). The scorer outputs per-frame progress percentages that are compared to timestamps via Spearman correlation. The experimental setup and metrics are described in Sec. 4; data and processing (Catrap playthroughs, clip extraction, anchor and shuffle) are detailed in the supplementary material. The supplement provides all hyperparameters.

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

# A  APPENDIX

## A.1  EXPERIMENTAL DETAILS

| Hyperparameter | Values |
|---|---|
| Env.steps | 57600 |
| Learning Rate (init $\rightarrow$ final) | 1e-5 $\rightarrow$ 5e-7 |
| Scheduler | cosine |
| GAE $\lambda_g$ | 0.95 |
| $\gamma_g$ | 0.99 |
| Value Loss Coeff. | 0.15 |
| KL $\beta$ | 0.05 |
| Policy Freeze (steps) | 2 |
| Grad Accum. Steps | 32 |
| Mini-batch Size | 1 |
| PPO Epochs | 2 |
| Obs. Image Length | 5 |
| Rollout Size | 256 |
| Max Episode Steps | 64 |
| Temperature | 0.2 |

Table 5: **Stage 2 (VL-DAC) hyperparameters.** Policy learns from the windowed *progress–time* $\rho \in [-1, 1]$ computed every $N$ steps (default $N = 15$, $L = 4$). $\gamma_g, \lambda_g$ are GAE parameters; LR follows a cosine decay.

**Hyperparameters (summary). Stage 1:** GRPO on a tuned VLM; the first frame is anchored, and the remaining $K-1$ frames are shuffled; the listwise reward is $R = \text{spr}(\mathbf{p}, \mathbf{q})$, where $\mathbf{p}$ denotes *progress percentages* and $\mathbf{q}$ denotes temporal indices (timestamps); a moving-average baseline is used. **Stage 2:** Reward frequency $N = 15$; $L = 3$ shuffled-window evaluations per query; window length $m$ as in Sec. 3.3 (we ablate $N$ and $L$); per-episode standardization is used only for advantage estimation. The policy and value networks in Stage 2 are trained with VL-DAC (Bredis et al., 2025) and a multi-step variant of GRPO, LOOP (Chen et al., 2025). Importantly, no environment rewards are used at any stage. Additionaly we employ kl control on stage1, we set target kl to $0.1$. Also we list hyperparameters for stage1 and stage2 in tables 6 5 7

| Hyperparameter | Values |
|---|---|
| Algorithm steps | 225 |
| Learning Rate (init $\rightarrow$ final) | 1e-5 $\rightarrow$ 1e-6 |
| Scheduler | linear with warmup |
| Num. warmup steps | 10 |
| KL $\beta$ | 0.05 |
| Grad Accum. Steps | 32 |
| Mini-batch Size | 1 |
| PPO Epochs | 1 |
| Obs. Image Length | 5 |
| Rollout Size | 120 |
| $\tau$ threshold | 0.5 |
| $\tau$ threshold steps | 3 |
| K | 4 |
| Temperature | 1 |

Table 6: **Stage 2 (LOOP) hyperparameters.** Multi-step GRPO variant using the same correlation-only reward. Training optionally applies *starting-point refreshing* when reward exceeds threshold $\tau$ to enable long-horizon completion.

| Hyperparameter | Values |
|---|---|
| Algo steps | 200-400 |
| Learning Rate | 1e-5 |
| Scheduler | constant with warmup |
| Num. warmup steps | 10 |
| Grad Accum. Steps | 16 |
| Mini-batch Size | 1 |
| GRPO Epochs | 1 |
| Obs. Image Length | 15 |
| K | 4 |
| Temperature | 1 |

Table 7: **Stage 1 scorer hyperparameters.** GRPO on anchor+shuffle clips; the VLM predicts per-frame *progress percentages* and maximizes *progress–time* Spearman $\rho$. After convergence, the scorer is frozen for Stage 2.

## A.2    ABLATIONS

In this section, we present an expanded set of ablations evaluating both algorithmic hyperparameters of RTA and architectural choices for the downstream policy. These studies are designed to isolate the contribution of each component and determine the robustness of RTA across implementation variations. All experiments are conducted on Catrap environments and results are averaged over multiple seeds.

**Policy backbone choice**. We further evaluate how RTA interacts with different policy backbones, comparing a VLM with a lightweight MLP trained with PPO. Table 8 summarizes the results for two downstream settings: providing RTA reward every $N = 15$ steps and providing reward only at the end of an episode.

We observe that the optimal schedule differs between the VLM and MLP policies, plausibly due to different exploration dynamics: an initialized MLP benefits from termination step RTA feedback (we reward full trajectory), whereas the VLM policy often benefits from shorter-window shaping. Overall, these results are consistent with the view that RTA's gains are driven by the reward signal's ordering stability rather than a particular policy architecture.

**Window mechanics**. We vary the window length $m$ around our default. Across Catrap levels, we find that performance is stable with the default value $m = 15$ and higher, while very short windows (too few frames) perform worse, as expected. For larger windows than default, convergence tends

| Task | Level 2 | Level 4 | Level 6 |
|---|---|---|---|
| VLM + RTA (reward every 15 steps) | 1.00 ± 0.00 | 0.60 ± 0.28 | 0.33 ± 0.34 |
| VLM + RTA (only-end reward) | 0.33 ± 0.19 | 0.00 ± 0.00 | 0.00 ± 0.00 |
| MLP + RTA (reward every 15 steps) | 0.79 ± 0.14 | 0.27 ± 0.21 | - |
| MLP + RTA (only-end reward) | 1.00 ± 0.00 | 1.00 ± 0.00 | 1.00 ± 0.00 |

Table 8: **Downstream PPO evaluation results. Mean ± std over 3 seeds for VLM-based agents and 5 seeds for MLP agents.**

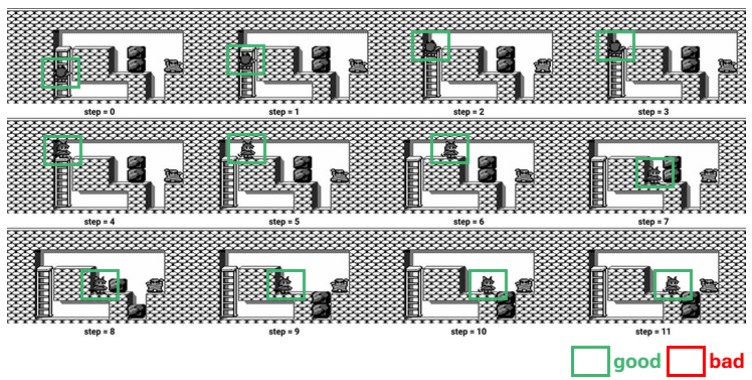

Figure 6: Example of good level compeletion with well-trained stage 1 model with reward 0.77 to be the same, and inference is slower due to the larger context processed by the reward model. We fix the reward frequency to 15 and $L$ to 2.

**Reward frequency**. We vary the frequency at which reward is provided to the learning process. We find that this parameter does not qualitatively affect stability, and its effect on convergence speed appears to depend more on optimization dynamics. We fix the window length to 15 and $L$ to 2.

**Number of shuffles L**. Reducing the number of shuffles to 1 degrades reward stability and can lead to slower convergence, indicating that multiple shuffles are important for learning a robust ordinal signal. We fix the window length to 15 and reward frequency to 15.

More detailed results of hyperparameters sweep can be found in 9.

| Ablation: Window Length $M$ | Success rate | Update steps |
|---|---|---|
| 5 | 0.73 ± 0.09 | 65.33 ± 12.76 |
| 15 | **1.00 ± 0.00** | **37.00 ± 10.98** |
| 25 | **1.00 ± 0.00** | 43.33 ± 5.79 |
| **Ablation: Reward Frequency** | | |
| 5 | **1.00 ± 0.00** | 175.60 ± 76.53 |
| 15 | **1.00 ± 0.00** | **37.00 ± 10.98** |
| 25 | **1.00 ± 0.00** | 128.67 ± 84.32 |
| **Ablation: Number of Shuffles $L$** | | |
| 1 | **1.00 ± 0.00** | 143.66 ± 63.43 |
| 2 | **1.00 ± 0.00** | **37.00 ± 10.98** |
| 4 | 0.93 ± 0.09 | 123.33 ± 83.83 |

Table 9: **Ablation study on window length $M$, reward frequency, and number of shuffles $L$.**

### A.3 EXAMPLES

**Level 3 performance** Notably, level 3 did not converge within 200 steps, prompting further experiments with longer training on this level (see Fig. 10).

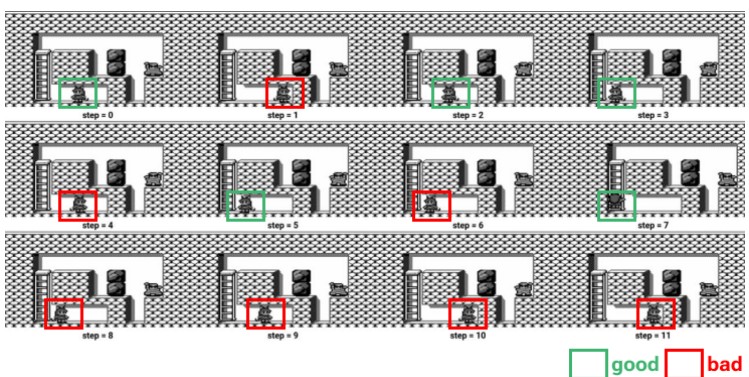

Figure 7: Example of bad level compeletion with well-trained stage 1 model with reward -0.04

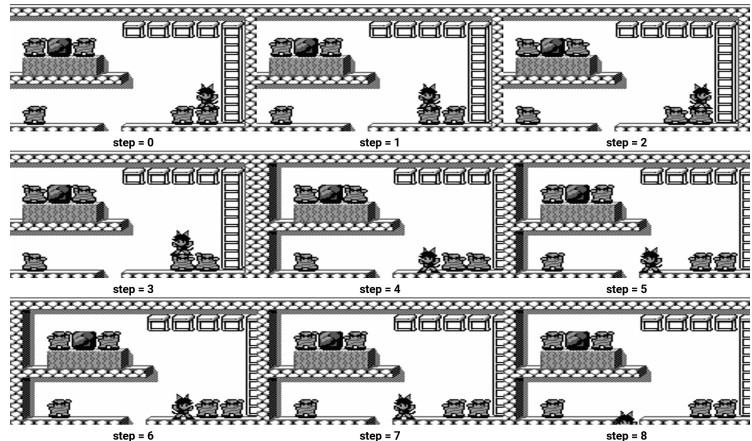

Figure 8: Example of bad level compeletion with undertrained stage 1 model with reward 0.66 (high)

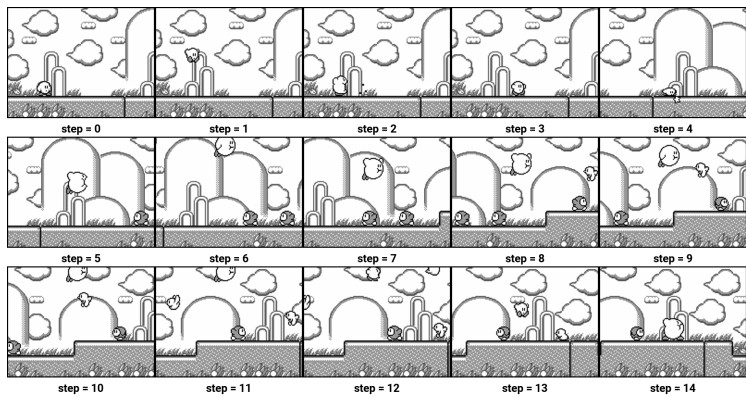

Figure 9: Example of good level compeletion with undertrained stage 1 model with reward 0.15 (low)

**Use of language models.** We used large language models solely for *editorial* assistance: light copy-editing (grammar, style, typography), tightening and clarifying prose, and making figure/table captions self-contained. LLMs were *not* used to design experiments, select hyperparameters, generate results, or fabricate analyses; all numbers, plots, and tables come from our code and logs. Any LLM-edited passages were reviewed and, when technical content was involved, rewritten by the authors. Suggested references (if any) were verified against the cited sources. No code or data was produced by LLMs.

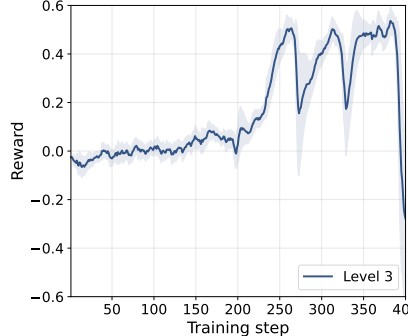

Figure 10: **Level 3 exposes training sensitivity.** Over 400 GRPO steps, *progress–time* $\rho$ alternates between gains and dips, indicating that ambiguous content learns but benefits from careful data prep and sufficient duration.

