# OpenReview forum: "Rank-Then-Act: Reward-Free Control from Frame-Order Progress"
_ICLR.cc/2026/Conference — Submitted to ICLR 2026_

### Official Review · Reviewer_UiJZ · 2025-10-22

**Soundness:** 1
**Presentation:** 3
**Contribution:** 2
**Rating:** 2
**Confidence:** 4

**Summary:**

This paper learns a ranking-based reward from expert demonstration videos by assigning progress scores to a shuffled set of k frames. They then use this model as a reward by calculating the Spearman rank correlation between the predicted and ground truth orderings, and train a VLM-based policy.

**Strengths:**

The method of learning a progress reward by using the Spearman rank correlation between the progress and ground truth frame scores is an original and distinct way of learning a reward function from expert videos. I think that this paper distinguishes itself from other work that learns a progress reward in this regard by using a novel methodology. Their methods section and explanation were clear.

**Weaknesses:**

In order to verify the efficacy of their method, in the plots or tables that just illustrate their learned reward learning curve, I would like to see either a success rate or an oracle reward.  Including a comparison to a non-VLM ranking method (in the related works, it describes rank2reward [3] as similar) and a non-VLM-based policy class would help demonstrate the effectiveness and show the benefits of using a VLM.   In section 4.2, the authors claim that per-level training is more robust, which is also shown in Figure 4.  Improved generalization capabilities, as described in [2, 4], are one of the advantages of using a VLM as a base model,  so I would like to understand the benefits a VLM brings to this work.

This paper compares against two baselines. In Figure 5 and Table 1, there is a comparison against a baseline model that lacks reward, which achieves 0% success, and a reward of 0 under their learned reward function.  Based on a correlation of 0, I suspect that the baseline is selecting random actions, but I would like more details about what this baseline is.  In Table 1, they also compare expert-supervised fine-tuning to RTA (their method). Including the other five levels, providing more detail about the SFT baseline, and reporting success rates instead of learned rewards would make this comparison clearer.

For the expert SFT, they augment a dataset of 200 trajectories with chain of thought reasoning, and fine-tune their VLM on this dataset. I wonder what the effects of chain of thought reasoning are on this SFT policy performance. I think comparing against a smaller policy class, given the single-task nature of this problem formulation, or a VLA, which is the standard way to adapt a VLM to a policy [1, 2], would help demonstrate the strengths of this method.

The authors also claim that other inverse RL methods have not been adapted to the vision-language setting at scale. They do not use language conditioning in a goal conditioning way (as mentioned in section 3.1), and inverse RL methods have been robustly applied to image-based robotic manipulation, as well as Atari-style games [3, 4, 5]. I also think a comparison to a sparse reward DQN would help show the performance of RTA.

The paper also claims that they do not require any task-specific supervision, but this method does require expert demonstration videos on a per-task basis.

[1] Kevin Black, Noah Brown, Danny Driess, et al. (2024). Pi0: A Vision-Language-Action Flow Model for General Robot Control.

[2] Moo Jin Kim, Karl Pertsch, Siddharth Karamcheti, et al. (2024). OpenVLA: An Open-Source Vision-Language-Action Model.

[3] Daniel Yang, Davin Tjia, Jacob Berg, et al. (2024).  Rank2Reward: Learning Shaped Reward Functions from Passive Video.

[4] Yecheng Jason Ma, Joey Hejna, Ayzaan Wahid, et al. (2024) Vision Language Models are In-Context Value Learners.

[5] Aaron Tucker, Adam Gleave, Stuart Russell. (2018). Inverse reinforcement learning for video games.

**Questions:**

Why in Table 1 does the SFT baseline, when it doesn’t beat RTA, have a high negative correlation?  It appears to have a higher magnitude correlation (either positive or negative) than RTA across the three levels.

What are the reasons for using a VLM as the base policy class and the base model for the learned reward function? I would also be curious to see the effects of including vs not including the reasoning traces.

Why only score every N steps instead of at every step during policy training?

---

> ### Author Response · Authors · 2025-11-26
>
> In response to your concerns, we (i) broadened Stage-2 evaluation to Catrap L2/L4/L6 and Kirby level-0, (ii) added an oracle sparse-reward comparison with matched policy, optimizer, and training budget, (iii) implemented stronger video-only baselines—Rank2Reward, VLM-RM (and a regularized variant), and an in-context VLM (“GVL-style”) scorer, using the same demonstrations and rollout budgets for fair comparison, (iv) clarified the “no-reward” and SFT baselines (including why SFT can exhibit negative correlation when moving away from demonstration ordering), and (v) expanded analysis of reward cadence by varying the scoring frequency reward.
>
> ### VLM based generalization
>
> W1 and Q2
>
> We use a VLM as the reward model because it generalizes better for frame-ordering across diverse video sources. To evaluate this, we trained RTA on (a) a single full-game Catrap playthrough, (b) a single full-game Kirby playthrough, and (c) a mixture of YouTube PyBoy playthroughs from multiple games and players (excluding Catrap and Kirby). These sources differ in visual appearance (compression artifacts, overlays, varied play styles) but share the same underlying game dynamics.
>
> Additionally, here and in experiments below we add experiments on harder Catrap levels 4 and 6 that require backtracking and have dead-ends, and we run experiments on Kirby game that require long horizon interaction. Kirby level-0 in VideoGameBench [1] as a challenging benchmark on which many frontier VLMs fail (reported 0% success for several large proprietary models). Stage-1 scorer training data varies; Stage-2 is trained on the listed level with the same pipeline.
>
> | Stage1 training data\ Stage 2 training level | L2 | L4 | L6 | Kirby-lvl0 |
> | --- | --- | --- | --- | --- |
> | Per-task | 1.00 ± 0.00 | 0.60 ± 0.28 | 0.33 ± 0.34 | 0.07 ± 0.09 |
> | Youtube | 1.00±0.00 | 0.47±0.25 | 0.60±0.28 | 0.20±0.16 |
> | Full Catrap | 1.00±0.00 | 0.47±0.38 | 0.60±0.28 | 0.07±0.09 |
> | Full Kirby | - | - | - | 0.07±0.09 |
>
> These results indicate that the same Stage‑1 scorer, trained on pooled or YouTube videos, can be reused across multiple Catrap levels and Kirby without per-level or per-game fine-tuning, while still supporting competitive downstream success. This demonstrates that RTA can tolerate visual and domain shifts in the expert videos. To further test robustness across VLM backbones, we also run RTA with **Qwen2.5-3B-VL** on Kirby level-0 and achieve 0.20±0.28 success, which hard for even closed-source frontier models [1].
>
> For Stage 2, we use a VLM-based policy as the actor in a discrete action environment, analogous to prior VLM applications in continuous domains [2, 4], which have demonstrated improved generalization to novel objects and environments [2, 3]. Additionally, in section Downstream Policy Evaluation, we also conduct experiments with an MLP backbone to verify that our method performs effectively with simpler policies.
>
> ### Comparison with oracle reward
>
> W1:
>
> To better contextualize the quality of the learned shaping reward, we compare RTA directly against an oracle sparse-reward baseline that has access to the environment’s ground-truth success signal. Both methods use the same policy architecture, optimization scheme, and training budget. Table 1 summarizes success rates (mean ± std over three seeds):
>
> | Reward | L2 | L4 | L6 | Kirby-lvl0 |
> | --- | --- | --- | --- | --- |
> | RTA (ours) | 1.00 ± 0.00 | 0.60 ± 0.28 | 0.33 ± 0.34 | 0.07 ± 0.09 |
> | Oracle reward | 0.50 ± 0.22 | 0.07 ± 0.09 | 0.20 ± 0.00 | 0.40 ± 0.28 |
>
> Across Catrap levels (L2, L4, L6), RTA consistently outperforms direct optimization of the sparse oracle reward (under a matched training budget), suggesting that correlational progress provides a more informative and exploration-friendly learning signal for VLM-based policies. On Kirby level-0, however, the oracle reward remains stronger, likely due to the task’s extended horizon and substantial backtracking requirements. We now explicitly highlight this as a limitation and discuss potential extensions - such as multi-scale temporal windows and hierarchical progress modeling - that could improve long-horizon robustness.

---

> ### Author Response · Authors · 2025-11-26
>
> ### Comparison with oracle reward
>
> W1
>
> To better contextualize the quality of the learned shaping reward, we compare RTA directly against an oracle sparse-reward baseline that has access to the environment’s ground-truth success signal. Both methods use the same policy architecture, optimization scheme, and training budget. Table 1 summarizes success rates (mean ± std over three seeds):
>
> | Reward | L2 | L4 | L6 | Kirby-lvl0 |
> | --- | --- | --- | --- | --- |
> | RTA (ours) | 1.00 ± 0.00 | 0.60 ± 0.28 | 0.33 ± 0.34 | 0.07 ± 0.09 |
> | Oracle reward | 0.50 ± 0.22 | 0.07 ± 0.09 | 0.20 ± 0.00 | 0.40 ± 0.28 |
>
> Across Catrap levels (L2, L4, L6), RTA consistently outperforms direct optimization of the sparse oracle reward (under a matched training budget), suggesting that correlational progress provides a more informative and exploration-friendly learning signal for VLM-based policies. On Kirby level-0, however, the oracle reward remains stronger, likely due to the task’s extended horizon and substantial backtracking requirements. We now explicitly highlight this as a limitation and discuss potential extensions - such as multi-scale temporal windows and hierarchical progress modeling - that could improve long-horizon robustness.
>
> ### SFT clarification
>
> W2, W3 and Q1
>
> The baseline in Table 1 is the default Qwen2.5-VL-7B-Instruct without any additional training, it did not solve the level in any run. For the SFT baseline, we collected 200 chain-of-thought trajectories by prompting gpt-o3 to generate reasoning traces for hand-collected images of the perfect trajectory. These traces were then used during fine-tuning. SFT achieves zero success on levels 2 and 6, but always solves level 4. Where it fails (L2 and L6), the agent’s actions often move it away from “later” demonstration states, resulting in negative correlations. On the level it solves reliably (L4), the correlation is positive.
>
> For wider baseline comparison check next section.
>
> ### Baseline comparison
>
> W1, W2 and W4
>
> We include a broader and better-matched set of baselines that follow the same paradigm of learning a scalar score from demonstration videos and using this score as reward within the same video-only setting:
>
> - Rank2Reward [5]: CNN-based ranker trained on passive clips.
> - VLM-RM [6]: CLIP-style reward model scoring observations against goal text.
> - VLM-RM (reg., α = 0.5) [6]: A regularized version interpolating between zero-shot and fine-tuned reward for improved calibration.
> - GVL-style baseline [3]: as the scorer, we use the Stage-1 model without additional training.
>
> All methods use exactly the same expert videos, observation/action spaces, and rollout budgets to ensure fair comparisons. Table 2 reports maximum success under matched conditions (mean ± std over three seeds):
>
> | Method | L2 | L4 | L6 | Kirby-lvl0 |
> | --- | --- | --- | --- | --- |
> | RTA (ours) | 1.00 ± 0.00 | 0.60 ± 0.28 | 0.33 ± 0.34 | 0.07 ± 0.09 |
> | Rank2Reward | 0.60 ± 0.28 | 0.20 ± 0.00 | 0.13 ± 0.09 | – |
> | VLM-RM | 0.40 ± 0.16 | 0.00 ± 0.00 | 0.07 ± 0.09 | 0.00 ± 0.00 |
> | VLM-RM (reg., α=0.5) | 0.27 ± 0.19 | 0.00 ± 0.00 | 0.00 ± 0.00 | 0.00 ± 0.00 |
> | GVL | 0.47 ± 0.25 | – | – | – |
>
> Qualitatively, Rank2Reward and VLM-RM are less stable when the policy deviates from demonstration trajectories or when the goal text under-specifies task structure. In contrast, RTA’s rank-based reward remains informative under moderate off-trajectory exploration, yielding higher success across the evaluated Catrap levels.
>
> To analyze reward–success alignment, we measure the Spearman correlation between cumulative reward and binary episode success. For each method–level pair, we select the best-performing seed based on validation success. When a method never succeeds on a level, correlation estimates are not meaningful; we report these as 0 for completeness.
>
> | Method | L2 | L4 | L6 | Kirby-lvl0 |
> | --- | --- | --- | --- | --- |
> | RTA (ours) | 0.76 | 0.48 | 0.42 | 0.13 |
> | Rank2Reward | −0.04 | −0.02 | −0.04 | – |
> | VLM-RM | 0.53 | 0.00 | −0.33 | 0.00 |
> | VLM-RM (reg., α=0.5) | 0.25 | 0.00 | 0.00 | 0.00 |
> | GVL | −0.01 | – | – | – |
>
> RTA exhibits strong positive alignment (ρ ≈ 0.4–0.8) across Catrap levels, whereas other baselines show little or inconsistent alignment. On Kirby-lvl0, all methods, including RTA, show weaker alignment (ρ ≈ 0.13), consistent with the challenges of long-horizon tasks. Also we attach visual examples of [good](https://8upload.com/display/5568e79f50b4075f/Frame_12.png.php) and [bad](https://8upload.com/display/37e6f5f927fc05b7/Frame_11.png.php) (in terms of success) level completion and their corresponding correlation rewards 0.77 and -0.04.

---

> > ### Author Response · Authors · 2025-11-26
> >
> > ### Downstream policy evaluation
> >
> > W1, W3, W4, Q3: Lack of baseline comparisons in policy evaluation.
> >
> > A: To remove confounding factors introduced by VLM-based agents, we conducted additional experiments using a standard PPO agent with an MLP backbone. Under this uniform pipeline, we run RTA on Catrap L2 and L4 with various reward-giving strategies.  Reward every N steps is working better for VLM-based approach, while stage 1 reward only on termination step working better for MLP. This happen due different exploration/exploitations of models, randomly initiliazed MLP requires long-trajectory reward, while VLM that already can move smoothly or stall works better with shorter window reward. Comparison can be found in the table below
> >
> > **Downstream PPO evaluation results:**
> >
> > |  | L2 | L4 |
> > | --- | --- | --- |
> > | VLM + RTA reward every N steps | 1.00 ± 0.00 | 0.60 ± 0.28 |
> > | VLM + RTA Only-end reward | 0.33 ± 0.19 | 0.0 ± 0.0 |
> > | MLP + RTA reward every N steps  | 0.79 ± 0.14 | 0.27± 0.21 |
> > | MLP + RTA Only-end reward | 1.00 ± 0.00 | 1.00 ± 0.00 |
> >
> > Table 3 further indicates that RTA provides a sufficiently informative reward signal for an MLP-based PPO policy to learn after enough interaction steps, suggesting the gains mainly come from the stability and ordering consistency of the proposed scorer rather than from VLM-specific architectural or optimization choices. Since DQN is an off-policy algorithm, we use PPO for these comparisons to keep the training setup on-policy and algorithmically matched across methods.
> >
> > ### Ablation study:
> >
> > Q3
> >
> > We expanded the ablation studies to analyze the effect of reward-update frequency. Because dense reward computation requires multiple forward passes through a pretrained VLM, providing feedback at every step is computationally expensive. We therefore examine reward frequencies of 5, 15, and 25 steps (window length fixed to 15, with L=2)
> >
> > | reward frequency | Success rate | Steps to achieve maximum success rate |
> > | --- | --- | --- |
> > | 5 | 1.00 ± 0.00 | 175.6 ± 76.53  |
> > | 15 | 1.00 ± 0.00 | 37.0 ± 10.98  |
> > | 25 | 1.00 ± 0.00 | 128.67 ± 84.32 |
> >
> > The algorithm remains stable across frequencies and ultimately reaches full success on Catrap L2. We highlight that the choice mainly reflects computational trade-offs rather than qualitative differences in learning dynamics.
> >
> > ### References
> >
> > [1] VideoGameBench: Can Vision-Language Models complete popular video games?
> >
> > [2] OpenVLA: An Open-Source Vision-Language-Action Model
> >
> > [3] Vision Language Models are In-Context Value Learners
> >
> > [4] Pi0: A Vision-Language-Action Flow Model for General Robot Control.
> >
> > [5] Rank2Reward: Learning Shaped Reward Functions from Passive Video
> >
> > [6] Vision-Language Models are Zero-Shot Reward Models for Reinforcement Learning

---

> > > ### Author Response · Authors · 2025-11-26
> > >
> > > We hope our revisions and clarifications address your concerns and would appreciate your reconsideration of the overall score. If anything remains unclear, we are ready to provide further explanation.

---

### Official Review · Reviewer_jdVY · 2025-10-31

**Soundness:** 1
**Presentation:** 3
**Contribution:** 2
**Rating:** 2
**Confidence:** 5

**Summary:**

This paper proposes a two-stage approach for control without environment rewards. Stage 1 trains a VLM scorer on shuffled expert demonstration frames to predict progress percentages via listwise ranking (GRPO) with an anchor–shuffle mechanism. Stage 2 freezes the scorer and uses Spearman correlation between predicted progress and temporal indices over sliding windows as the reward signal for PPO-based policy learning. Evaluated primarily on PyBoy Catrap.

**Strengths:**

- Simple, implementable correlation-based reward signal.

- Anchor–shuffle mechanism to reduce trivial temporal shortcuts.

- Clear method description with hyperparameter tables; the pipeline is easy to reproduce in principle.

**Weaknesses:**

- Evidence too weak to justify publication. The paper reports results from only 2 seeds, EMA-smoothed, with no error bars, confidence intervals, or per-seed reporting. This violates basic standards for RL research. Evaluation for policy learning is limited to a single game (Catrap): while the progress scorer is illustrated on a few other games, the end-to-end control results (Stage-2 PPO with the learned reward) are reported only on Catrap. No Atari, continuous control, robotics, or even other PyBoy games for policy training. Additionally, no ablations on critical design choices: window size (m), query frequency (N), number of shuffles (L), temperature, anchor vs. no-anchor, or robustness to parsing failures (regex match -> R_min = -1). The LOOP variant uses “starting-point refreshing” (a save-state curriculum from best states), which isn’t standard RL protocol and appears necessary for success; the paper’s own figure suggests reward can rise without success unless refreshing is used. Required to meet the empirical bar: >=5 seeds with proper statistical analysis, broader task coverage, and principled ablations.

- Related-work omissions materially overstate novelty. The paper claims to be "the first approach that enables a VLM to learn control policies in a fully reward-free environment." That claim is false: prior work already uses VLMs (e.g., CLIP-based reward models) to train policies with no environment rewards. In addition, the key ingredients have prior art: (i) VLMs as zero-shot reward models for control without env rewards (VLM-RM from Rocamonde et al.), (ii) learning progress/value by ordering shuffled frames with VLMs (GVL from Ma et al.), (iii) extracting shaped rewards from passive video via temporal ranking (Rank2Reward from Yang et al.), and (iv) temporal order as self-supervision (Shuffle & Learn from Misra et al.). The specific combination of Spearman correlation over windowed predictions may be novel, but claiming "first" while omitting comparisons to these directly relevant methods significantly overstates the contribution. There are no baselines against VLM-RM (CLIP-style rewards), Rank2Reward, T-REX/D-REX, or XIRL/ORCA-style temporal alignment methods.

- Brittle core assumption (monotonic progress). The method presumes progress increases with time inside successful demos: r_t ∝ ρ(predicted_progress[W_t], time_index[W_t]), where W_t is a sliding window. This assumption is frequently false in multi-stage manipulation (staging, re-grasps, approach–retreat) and puzzle games (backtracking, exploration). The paper neither (i) justifies the assumption’s scope, (ii) stress-tests on backtracking tasks, nor (iii) validates that $\rho$ correlates with actual task success rather than demo tempo or superficial local coherence. The reward measures agreement with a particular demonstration’s temporal ordering, not task progress per se. A direct episodic analysis of corr($\rho$, return) is needed.

- Non-Markovian reward without proper treatment. The reward at time t depends on a window W_t = (s_{t−m+1}, …, s_t), i.e., on history the policy may not observe, while PPO is run as if the setting were standard MDP RL. At minimum, the paper should acknowledge the theoretical mismatch and either (i) include the entire reward window in the observation or use recurrence to restore effective Markovianness, or (ii) provide empirical justification that the mismatch does not harm training. As written, the assumptions behind the PPO setup are not satisfied.

**Questions:**

The following requests correspond to unanswered questions in the paper which I believe must be addressed to clear the bar for publication.

- Provide ≥5 seeds with proper statistical reporting (means, confidence intervals, and per-seed curves) for all main results.

- Broaden control evaluation beyond Catrap: include at least one additional PyBoy game for policy learning and a non-Atari continuous-control or manipulation domain to support generality claims.

- Add fair baselines with matched compute and data budgets against directly relevant methods, as judged by the authors (e.g. VLM-RM, Rank2Reward, T-REX/D-REX, and XIRL/ORCA-style temporal alignment).

- Improve ablations. Window mechanics: vary window size m and query frequency N; compare stacking enough frames so the observation covers the reward window vs. adding recurrence; report sensitivity. Anchor–shuffle ablations: anchor vs. no-anchor; number of shuffles L; temperature; show effects on both scorer quality and policy performance.

- Quantify how well the proposed reward aligns with task success: report episode-wise correlation/calibration between the ρ-based reward and returns/success across levels.

---

> ### Author Response · Authors · 2025-11-25
>
> ### Summary of updates
>
> In response to your concerns, we (i) broadened Stage-2 evaluation to Catrap L2/L4/L6 + Kirby level-0, (ii) implemented Rank2Reward, VLM-RM (and a regularized variant), and a GVL-style baseline with matched data and compute, (iii) added window-mechanics and shuffle ablations, (iv) quantified reward–success alignment and compared against an oracle sparse-reward baseline. Also we will revise the text to better reflect the current results and related literature.
>
> ### Statistical significance
>
> W1, Q1, and Q2: Weak statistical justification and scope
>
> **Harder environements:**
>
> We ran all main Stage-2 experiments with 3 random seeds (up from 2), and we now report mean ± standard deviation of success rates across seeds. Additionally, we added experiments on harder Catrap levels L4 and L6, which require backtracking and contain dead-ends, and we ran experiments on the Kirby game, which requires long-horizon interaction.
>
> Under our reward-free protocol (no environment reward, only progress–time correlation), RTA achieves the following mean ± std of best-achieved success over training (averaged over 3 seeds):
>
> |  | Catrap lvl2 | Catrap lvl4 | Catrap lvl6 | Kirby-lvl0 |
> | --- | --- | --- | --- | --- |
> | RTA (ours) | 1.00±0.00 | 0.60±0.28 | 0.33±0.34 | 0.07±0.09  |
>
> Kirby level-0 is identified in VideoGameBench [1] as a challenging benchmark on which many frontier VLMs fail (reported 0% success for several large proprietary models). Catrap L4 and L6 introduce irreversible ‘dead-end’ states, so the agent must reason about exploration and backtracking rather than follow a single monotone path. In contrast, RTA obtains a mean success of 0.07±0.09 despite using only video-based progress signalss. We note that 5-seed experiments are compute-heavy in the LLM/VLM setting, and that 3-seed reporting is common in recent LLM/VLM agent work [5, 8].
>
> **VLM backbone:**
>
> To further test robustness across VLM backbones, we also ran RTA with Qwen2.5-3B-VL on Kirby level-0 and achieved 0.20±0.28 success, which is challenging even for closed-source frontier models [1].
>
> **Generalization:**
>
> To examine generalization of the Stage-1 scorer across video sources and to empirically support this design choice, we carried out an additional experiment where the Stage-1 scorer is trained on (a) a single full-game Catrap playthrough and (b) a mixture of YouTube PyBoy playthroughs from multiple games (excluding Catrap and Kirby). These sources differ in visual appearance (compression artifacts, overlays, distinct play styles, etc.) but share the same underlying game dynamics.
>
> |  | L2 | L4 | L6 | Kirby-lvl0 |
> | --- | --- | --- | --- | --- |
> | YouTube | 1.00±0.00 | 0.47±0.25 | 0.60±0.28 | 0.20±0.16 |
> | Full Catrap | 1.00±0.00 | 0.47±0.38 | 0.60±0.28 | 0.07±0.09 |
> | Full Kirby | – | – | – | 0.07±0.09 |
>
> These results show that the same Stage-1 scorer trained on pooled or YouTube videos can be reused across multiple Catrap levels and Kirby without any per-level or per-game fine-tuning, while still supporting competitive downstream success. This provides evidence that RTA can tolerate visual/domain shift in the expert videos.

---

> > ### Author Response · Authors · 2025-11-25
> >
> > ### Baseline comparison
> >
> > W2 and Q3: Lack of baseline comparison
> >
> > We implemented baselines that instantiate the same “train/use a model to score trajectories and use that score as reward” pattern in our video-only regime:
> >
> > - Rank2Reward [2]: a CNN-based ranker trained on sampled clips to produce a scalar shaped reward from passive videos.
> > - VLM-RM [3]: a CLIP-style reward model that scores observations against goal text, used as a dense reward.
> > - VLM-RM (reg., α=0.5) [3]: a regularized variant where we interpolate between zero-shot and fine-tuned reward to improve calibration.
> > - GVL-style baseline [4]: as the scorer, we use the Stage-1 model without additional training.
> >
> > All methods use exactly the same expert videos, observation/action spaces, and rollout budgets as RTA, so comparisons are matched in data and compute. Table 1 shows mean ± std of maximum success across training over 3 seeds:
> >
> > | Method | L2 | L4 | L6 | Kirby-lvl0 |
> > | --- | --- | --- | --- | --- |
> > | **RTA (ours)** | **1.00 ± 0.00** | **0.60 ± 0.28** | **0.33 ± 0.34** | **0.07 ± 0.09** |
> > | Rank2Reward | 0.60 ± 0.28 | 0.20 ± 0.00 | 0.13 ± 0.09 | – |
> > | VLM-RM | 0.40 ± 0.16 | 0.00 ± 0.00 | 0.07 ± 0.09 | 0.00 ± 0.00 |
> > | VLM-RM (reg., α=0.5) | 0.27 ± 0.19 | 0.00 ± 0.00 | 0.00 ± 0.00 | 0.00 ± 0.00 |
> > | GVL | 0.47 ± 0.25 | – | – | – |
> >
> > We observe that Rank2Reward and VLM-RM often fail to provide a stable shaping signal in our game-like tasks: their scalar rewards tend to saturate or become inconsistent when the agent visits states that deviate from the demonstration trajectories or the textual goal description. In contrast, RTA’s correlation-only reward, which depends only on the ordering of frames within a window, remains informative when the agent explores moderately off-demonstration states, leading to consistently higher success across levels.
> >
> > Regarding D-REX/XIRL/ORCA-style temporal alignment methods, we note that these methods operate in a different data regime: they require multiple demonstration trajectories of the same task, or additional structure such as subgoals. Adapting such methods would require augmenting the data (e.g., additional demonstrations or subgoal annotations) and thus have a different problem formulation. In this work, we therefore focus on baselines that are directly applicable in our regime (video-only reward models and rankers) and position D-REX/XIRL/ORCA as related but operating in a different data regime; we will state this explicitly in Related Work and Limitations.
> >
> > ### Ablation studies
> >
> > W1 and Q4
> >
> > We have substantially expanded the ablation study to cover the key design choices:
> >
> > - Window mechanics. We vary the window length m around our default. Across Catrap levels, we find that performance is stable with the default value m=15 and higher, while very short windows (too few frames) perform worse, as expected. For larger windows than default, convergence tends to be the same, and inference is slower due to the larger context processed by the reward model. We fix the reward frequency to 15 and n_repeats to 2.
> >
> > | window length M | Success rate | Update steps to achieve highest success rate |
> > | --- | --- | --- |
> > | 5 | 0.73 ± 0.09 | 65.33 ± 12.76 |
> > | 15 | 1.00 ± 0.00 | 37.0 ± 10.98 |
> > | 25 | 1.00 ± 0.00 | 43.33 ± 5.79 |
> > - Reward frequency. We vary the frequency at which reward is provided to the learning process. We find that this parameter does not qualitatively affect stability, and its effect on convergence speed appears to depend more on optimization dynamics. We fix the window length to 15 and n_repeats to 2.
> >
> > | reward frequency | Success rate | Update steps to achieve highest success rate |
> > | --- | --- | --- |
> > | 5 | 1.00 ± 0.00 | 175.6 ± 76.53 |
> > | 15 | 1.00 ± 0.00 | 37.0 ± 10.98 |
> > | 25 | 1.00 ± 0.00 | 128.67 ± 84.32 |
> > - Number of shuffles L. Reducing the number of shuffles to 1 degrades reward stability and can lead to slower convergence, indicating that multiple shuffles are important for learning a robust ordinal signal. We fix the window length to 15 and reward frequency to 15.
> >
> > | number of shuffles | Success rate | Update steps to achieve maximum success rate |
> > | --- | --- | --- |
> > | 1 | 1.00 ± 0.00 | 143.66 ± 63.43 |
> > | 2 | 1.00 ± 0.00 | 37.0 ± 10.98 |
> > | 4 | 0.93 ± 0.09 | 123.33 ± 83.83 |
> >
> > In the above experiments, an “update step” corresponds to refreshing the rollout buffer; each rollout buffer has size 64 environment steps. All ablations above were on Catrap L2 (3 seeds), using the same training budget, success is maximum evaluation success observed over training.
> >
> > Overall, these ablations suggest that RTA’s performance is not tuned to a single narrow hyperparameter setting.

---

> > > ### Author Response · Authors · 2025-11-25
> > >
> > > ### Reward design justification
> > >
> > > W4 and Q5
> > >
> > > To address this, we compute episode-wise Spearman correlation between cumulative RTA reward and binary episode success for each method and level. For each method–level pair, we take the best-performing seed (according to validation success) and compute the correlation across its evaluation episodes. When a method never solves a level, we cannot meaningfully estimate correlation; for completeness, we report these entries as 0 in the table, but they should be interpreted as “no evidence of positive alignment” in the current setup, likely due to task difficulty (e.g., backtracking and dead-ends).
> > >
> > > | Method | L2 | L4 | L6 | Kirby-lvl0 |
> > > | --- | --- | --- | --- | --- |
> > > | RTA (ours) | 0.76 | 0.48 | 0.42 | 0.13 |
> > > | Rank2Reward | −0.04 | −0.02 | −0.04 | – |
> > > | VLM-RM | 0.53 | 0.00 | −0.33 | 0.00 |
> > > | VLM-RM (reg., α=0.5) | 0.25 | 0.00 | 0.00 | 0.00 |
> > > | GVL | −0.01 | – | – | – |
> > >
> > > For Catrap levels, RTA’s cumulative correlation reward is strongly and positively aligned with success, whereas Rank2Reward and GVL show essentially no alignment, and VLM-RM exhibits unstable behavior (positive on L2, negative on L6). On Kirby level-0 the alignment is weaker (ρ ≈ 0.13), reflecting the much longer horizon and more frequent backtracking; we will highlight this in the paper as a limitation of our current setup on very long-horizon tasks. We also attach visual examples of [good](https://8upload.com/display/5568e79f50b4075f/Frame_12.png.php) and [bad](https://8upload.com/display/37e6f5f927fc05b7/Frame_11.png.php) (in terms of success) level completion and their corresponding correlation rewards (0.77 and −0.04).
> > >
> > > To further contextualize the reward, we compare RTA to an oracle sparse-reward baseline that has access to the environment’s ground-truth success signal. Table 2 reports success rates under the two reward schemes (same policy architecture, optimization, and budgets; mean ± std over 3 seeds):
> > >
> > > | Reward | L2 | L4 | L6 | Kirby-lvl0 |
> > > | --- | --- | --- | --- | --- |
> > > | RTA (ours) | 1.00 ± 0.00 | 0.60 ± 0.28 | 0.33 ± 0.34 | 0.07 ± 0.09 |
> > > | Oracle sparse | 0.50 ± 0.22 | 0.07 ± 0.09 | 0.20 ± 0.00 | 0.40 ± 0.28 |
> > >
> > > On Catrap levels L2, L4, and L6, RTA achieves higher success than optimizing the sparse oracle reward directly, suggesting that the correlation reward encourages a more effective exploration pattern for VLM policies than a purely terminal success signal. On Kirby level-0, the oracle reward remains stronger; we interpret this as evidence that our current instantiation of RTA is still challenged by very long-horizon platforming with extensive backtracking.
> > >
> > > Overall, these analyses provide direct evidence that, in the regimes we evaluate, the progress–time correlation reward is meaningfully aligned with task success and acts as an exploration-friendly surrogate.
> > >
> > > ### Acknowledgement of algorithm details
> > >
> > > W5: Non-Markovian reward / PPO assumptions not satisfied
> > >
> > > A: We clarify that we apply PPO as an approximate policy-gradient method in this POMDP setting, relying on the short observation stack to serve as a practical sufficient statistic for the reward. In practice, VLM/LLM-based policies operate under a truncated context window due to memory and token limits, so the policy never has access to the full sequence of states over which the reward is computed; this has been discussed in prior work [5, 6, 7]. Even with Markovian environment dynamics, the agent therefore experiences the problem as partially observed. Our windowed reward makes this mismatch explicit rather than introducing a qualitatively new issue.
> > >
> > > ### LOOP / starting-point refreshing (method clarification)
> > >
> > > W1: LOOP uses “starting-point refreshing” (save-state curriculum), which is non-standard and appears necessary for success.
> > >
> > > A: LOOP is an optional training variant that we included as a practical curriculum to improve sample efficiency in sparse, backtracking-heavy levels: periodically, the agent is reset from a set of previously reached high-progress save-states (“refreshing”) to reduce training complexity. Importantly, RTA does not rely on this curriculum: our main results and all baseline comparisons in Tables 1-2 use the standard PPO (applied to VLM) setup (no save-state refreshing).
> > >
> > > ### References
> > >
> > > [1] VideoGameBench: Can Vision-Language Models complete popular video games?
> > >
> > > [2] Rank2Reward: Learning Shaped Reward Functions from Passive Video
> > >
> > > [3] Vision-Language Models are Zero-Shot Reward Models for Reinforcement Learning
> > >
> > > [4] Vision Language Models are In-Context Value Learners
> > >
> > > [5] Reinforcement Learning for Long-Horizon Interactive LLM Agents
> > >
> > > [6] Fine-Tuning Large Vision-Language Models as Decision-Making Agents via Reinforcement Learning
> > >
> > > [7] Enhancing Vision-Language Model Training with Reinforcement Learning in Synthetic Worlds for Real-World Success
> > >
> > > [8] Group-in-Group Policy Optimization for LLM Agent Training

---

> > > > ### Author Response · Authors · 2025-11-25
> > > >
> > > > We hope our revisions and clarifications address your concerns and would appreciate your reconsideration of the overall score. If anything remains unclear, we are ready to provide further explanation.

---

### Official Review · Reviewer_Pa6W · 2025-11-04

**Soundness:** 2
**Presentation:** 2
**Contribution:** 2
**Rating:** 2
**Confidence:** 5

**Summary:**

This paper proposes a two-stage reward-free reinforcement learning framework that enables policy learning from expert video demonstrations without explicit task rewards. In Stage 1 (Rank), a vision–language model is fine-tuned using a GRPO objective to predict per-frame progress percentages from shuffled expert video clips. In Stage 2 (Act), the frozen progress scorer is used online to provide feedback: the Spearman correlation between predicted progress and environment timestamps over recent observation windows serves as the sole reward for policy optimization. Experiments on the PyBoy Catrap game demonstrate that RTA can train an agent to solve tasks using only video-based progress signals.

**Strengths:**

1. The paper explores a new formulation of reward-free control by leveraging rank correlation from expert video progress, removing the need for explicit rewards or adversarial imitation. The idea of using Spearman correlation of predicted progress as a dense reward is well-motivated.
2. The comparison with expert SFT baselines highlights the effectiveness of the progress-based rewards.

**Weaknesses:**

1. The overall design (training a model to score trajectories and using that score as a reward signal) resembles earlier works such as [A] and [B]. Although RTA’s reward shaping via video shuffle and Spearman correlation adds new intuition, it still assumes that short video clips encode the full policy value—a strong assumption that may not generalize to long-horizon or partially observable tasks.
2. The experiments are conducted only in a simplified simulation (PyBoy Catrap). Validation on more complex or real robotic control tasks would strengthen claims about generality. The paper would benefit from broader evaluation across standard RL benchmarks or real-world data.
3. Given that RTA relies on expert video data, comparisons with recent offline RL or inverse RL methods that use expert demonstrations are missing. Including such baselines (e.g., Rank2Reward [B]) would clarify the actual improvement over prior approaches.

Ref:

[A] A Vision-Language-Action-Critic Model for Robotic Real-World Reinforcement Learning.

[B] Rank2Reward: Learning Shaped Reward Functions from Passive Video.

**Questions:**

1. How does the proposed progress–time correlation reward perform when expert videos differ in the visual domain or embodiment (e.g., human videos vs. simulated robot views)? Would the scorer still generalize?
2. Could the authors extend RTA to handle long-horizon or hierarchical tasks, where local frame-order consistency does not imply overall progress?

---

> ### Author Response · Authors · 2025-11-21
>
> We thank the reviewer for the detailed feedback. In response, we (i) substantially broadened the evaluation to harder Catrap levels and the long-horizon Kirby level-0; (ii) implemented Rank2Reward, VLM-RM, and a GVL-style baseline with matched data and compute; (iii) added episodic reward–return correlation and oracle-reward comparisons to clarify the behavior of our progress signal; and (iv) evaluated Stage-1 scorer reuse across multiple games and YouTube video sources. Below we address each concern in turn.
> ### Broader scope
>
> W2: Experiments only in simplified simulation.
>
> A: Regarding the concern that our evaluation is limited to a simplified PyBoy setup (W2), we expanded the experimental suite to include harder levels of Catrap and a longer-horizon PyBoy Kirby task. Catrap L4 and L6 introduce irreversible ‘dead-end’ states, so the agent must reason about exploration and backtracking rather than follow a single monotone path. Kirby level-0, as identified in VideoGameBench [1], is a long-horizon platforming task on which many frontier VLMs fail to achieve any success (0% success rate across multiple attempts). In contrast, RTA obtains a mean success of 0.07±0.09 despite using only video-based progress signalss. All experiments show that model trained with Stage1 reward during Stage2 become capable of achieving success on each of above games and levels. Comparison of success rates of model you can find in table 1. All mean success rates and std’s are computed over 3 seeds.
>
> | Method | L2 | L4 | L6 | Kirby-lvl0 |
> | --- | --- | --- | --- | --- |
> | RTA(ours) | 1.00±0.00 | 0.60±0.28 | 0.33±0.34 | 0.07±0.09 |
>
> To further test robustness, we also run RTA with Qwen2.5-3B-VL on Kirby level-0, achieving 0.20±0.28 success, which is competitive with or better than reported performance of much larger closed-source models on this benchmark [1].
>
> We acknowledge that robotic and continuous-control benchmarks are a common setting for reward-modeling work; in this paper we deliberately focus on game environments and have therefore concentrated our additional experiments on substantially broadening the PyBoy suite.
>
> ### Comparison with baselines
>
> W3: Lack of comparison with Baselines
>
> A: In response to the request for comparisons with recent offline RL / IRL methods that also use expert demonstrations, we implemented Rank2Reward, a VLM-RM–style, and GVL in our PyBoy setting. These baselines are chosen to closely instantiate the ‘train a model to score trajectories and use that score as reward’ pattern, so that we can directly test whether RTA’s correlation-only reward provides a tangible advantage over these earlier designs in the same data regime. All methods are trained on exactly the same expert videos and evaluated under the same observation/action spaces and rollout budgets as RTA (same compute and time), so the comparison is matched in data and compute.
>
> Table 2 reports mean±std of maximum obtained success rates over all training over 3 seeds on Catrap Levels 2, 4, 6 and Kirby Level 0:
>
> |  | L2 | L4 | L6 | Kirby-lvl0 |
> | --- | --- | --- | --- | --- |
> | Ours (RTA) | 1.00±0.00 | 0.60±0.28 | 0.33±0.34 | 0.07±0.09 |
> | Rank2Reward | 0.60±0.28 | 0.20±0.0 | 0.13±0.09 | - |
> | VLM-RM | 0.40±0.16 | 0.0±0.0 | 0.07±0.09 | 0.0±0.0 |
> | VLM-RM (reg, \alpha=0.5) | 0.27±0.19 | 0.0±0.0 | 0.0±0.0 | 0.0±0.0 |
> | GVL | 0.47±0.25 | - | - | - |
>
> We observe that Rank2Reward (with a CNN-based scorer) and VLM-RM (goal-text–conditioned reward) often fail to provide a stable shaping signal on these game-like tasks: their learned rewards tend to saturate or become inconsistent when the agent visits states that deviate from the demonstration trajectories or learned representations. In contrast, RTA’s correlation-only reward, which depends only on the ordering of frames within a short window, remains informative even when the agent explores moderately off-demonstration states, leading to consistently higher success rates across levels.
>
> We will revise the introduction and related work to make the connections to previous work explicit and to more position our novelty better. Conceptually, RTA differs from previous approaches in terms of how it treat reward process. The new experiments above are designed precisely to show that, under the same video-only regime, this correlation-based reward yields more robust control than prior video-based methods.
>
> Additionally we note that we not directly compare our method to [2] because it concurrent work, but value paper contribution and will add it to related section.

---

> ### Author Response · Authors · 2025-11-21
>
> ### Method intuition
>
> W1 and Q2: Design justification and extension to long horizon tasks.
>
> A: We agree that our method implicitly assumes that short clips contain enough local information to decide whether the agent is making progress. This is analogous to a human viewer judging, from a 10–15 second segment of gameplay, whether the character is moving toward the goal. This assumption will not hold in all possible domains, notably, in heavily hierarchical tasks with extended backtracking, but in this setup we find it to be a reasonable approximation. Also we attach visual examples of [good](https://8upload.com/display/5568e79f50b4075f/Frame_12.png.php) and [bad](https://8upload.com/display/37e6f5f927fc05b7/Frame_11.png.php) (in terms of success) level completion and their corresponding correlation rewards 0.77 and -0.04.
>
> To provide empirical support, we report in Table 3 the episode-wise Spearman correlation between cumulative RTA reward and episode success for each method and level. For each method-level pair, we take the best-performing seed (selected by validation success) and compute correlation across its evaluation episodes. When a method never solves a level, the correlation is effectively undefined; in the table this appears as 0. If due to compute constraints we did not run experiment in specific setting such experiment in table appears as “-”.
>
> |  | L2 | L4 | L6 | Kirby-lvl0 |
> | --- | --- | --- | --- | --- |
> | Ours (RTA) | 0.76 | 0.476 | 0.42 | 0.13 |
> | Rank2Reward | -0.0389 | -0.0203 | -0.0359 | - |
> | VLM-RM | 0.53 | 0.0 | -0.33 | 0.0 |
> | VLM-RM (reg, \alpha=0.5) | 0.25 | 0.0 | 0.0 | 0.0 |
> | GVL | -0.011 | - | - | - |
>
> We additionally compare RTA to an oracle sparse-reward baseline that has access to the environment’s ground-truth success signal. Table 4 reports success rates under the two reward schemes (same policy architecture, optimization, and budgets, all mean success rates and std’s are computed over 3 seeds.):
>
> |  | L2 | L4 | L6 | Kirby-lvl0 |
> | --- | --- | --- | --- | --- |
> | Ours (RTA) | 1.00±0.00 | 0.60±0.28 | 0.33±0.34 | 0.07±0.09 |
> | Oracle | 0.50±0.22 | 0.07±0.09 | 0.2±0.0 | 0.40±0.28 |
>
> On Catrap levels L2, L4, and L6, RTA achieves higher success than optimizing the sparse oracle reward directly, suggesting that the correlation reward encourages more effective exploration for VLM policies than a sparse success signal. On Kirby Level 0, the oracle reward remains stronger; we interpret this as evidence that our current RTA setup is still challenged by very long-horizon platforming, and we now highlight this explicitly as a limitation. Overall, these results support the claim that RTA provides a practical exploration-friendly signal for VLM agents in the game-like tasks we consider, while also clarifying that very long-horizon tasks remain an open challenge.
>
> ### Generalization test
>
> A: To examine generalization of the Stage-1 scorer across video sources, we carried out an additional experiment where RTA is trained on (a) a single full-game Catrap playthrough, and (b) a mixture of YouTube PyBoy playthroughs from multiple games and players (not including Catrap and Kirby). These sources differ in visual appearance (compression artifacts, overlays, distinct play styles, etc.) but share the same underlying game dynamics. While we do not yet evaluate cross-embodiment generalization (e.g., human egocentric videos to robot states), we can test robustness to moderate visual and stylistic shifts within the same family of games.
>
> |  | L2 | L4 | L6 | Kirby-lvl0 |
> | --- | --- | --- | --- | --- |
> | Youtube | 1.00±0.00 | 0.47±0.25 | 0.60±0.28 | 0.20±0.16 |
> | Full Catrap | 1.00±0.00 | 0.47±0.38 | 0.60±0.28 | 0.07±0.09 |
> | Full Kirby | - | - | - | 0.07±0.09 |
>
> These results show that the same Stage-1 scorer trained on pooled or YouTube videos can be reused across multiple Catrap levels and Kirby without any per-level or per-game fine-tuning, and still supports competitive downstream success. This provides evidence that RTA can tolerate moderate visual/domain shift in the expert videos.
>
> [1] VideoGameBench: Can Vision-Language Models complete popular video games?
>
> [2] A Vision-Language-Action-Critic Model for Robotic Real-World Reinforcement Learning.

---

> > ### Author Response · Authors · 2025-11-22
> >
> > We hope our revisions and clarifications address your concerns and would appreciate your reconsideration of the overall score. If anything remains unclear, we are ready to provide further explanation.

---

> > > ### Author Response · Authors · 2025-11-26
> > >
> > > Dear reviewer,
> > > As the rebuttal period is approaching its end, we want to ensure that the clarifications and additional experiments we provided adequately address the concerns raised in the initial review. If any points remain unresolved or if further detail would support an accurate evaluation of the submission, we are ready to respond while the discussion phase is still open.
> > > Thank you again for your time and careful consideration.

---

### Official Review · Reviewer_mxSc · 2025-11-04

**Soundness:** 2
**Presentation:** 2
**Contribution:** 2
**Rating:** 4
**Confidence:** 3

**Summary:**

This paper introduces Rank-Then-Act framework, which uses a progress-percentage signal derived from expert video demonstrations. The framework trains a Vision–Language Model (VLM) progress scorer offline with a Group Relative Policy Optimization (GRPO) objective, assigning progress percentages to shuffled frames from expert gameplay. The progress percentage is used to provide feedback during training RL policy. On the PyBoy Catrap environment, RTA enables a VLM-based agent to solve levels using only expert videos, without any reward engineering.

**Strengths:**

1. Learning control directly from pixels without extrinsic rewards remains a significant challenge for the research community. This work is both timely and impactful, presenting a simple yet effective solution that successfully addresses several key technical obstacles.

2. Through extensive experiments in PyBoy environments, the authors empirically demonstrate that the proposed rank-correlation signal, i.e., derived from progress-percentage predictions, enables agents to solve tasks entirely without relying on environment rewards.

**Weaknesses:**

1. **Lack of reasoning, justification, and supporting evidence in the introduction:** The paper aims to develop a robust method for predicting progress signals from expert video snippets to facilitate VLM-based policy learning. However, the introduction does not sufficiently establish the *motivation* or *necessity* for this approach. Specifically, the authors should (1) highlight the key limitations of existing methods, (2) provide clear reasoning on why these limitations hinder achieving the desired objectives, and (3) present supporting evidence or analysis to justify the proposed solution. As it stands, the introduction primarily summarizes *what* the paper does, rather than why the approach is needed or how it advances the field.

2. **Missing baseline comparisons for the proposed progress scorer:** One of the central contributions of this work is the proposed *progress scorer*. However, the experimental section does not include direct comparisons with existing progress prediction or reward modeling methods. It remains unclear how the proposed scorer performs relative to prior approaches such as:
    - Ma et al., Vision-Language Models Are In-Context Value Learners, ICLR 2025
    - Hung et al., VICtoR: Learning Hierarchical Vision–Instruction Correlation Rewards for Long-Horizon Manipulation, ICLR 2025
    - Rocamonde et al., Vision–Language Models Are Zero-Shot Reward Models for Reinforcement Learning, ICLR 2024

A side-by-side quantitative and qualitative comparison is necessary to substantiate the claimed improvement.

3. **Lack of baseline comparisons in policy evaluation:** The authors should also evaluate the downstream policy learning performance by re-implementing or adapting the above baselines within the same framework. Without this comparison, it is difficult to determine whether the observed improvements stem from the proposed progress scorer or from other design factors in the training pipeline.

4. **Insufficient experimental validation of the primary contribution:** Prior works [1–3] have conducted extensive experiments using their respective progress or reward models across diverse tasks. To convincingly establish the first claimed contribution, the authors should provide a more comprehensive evaluation covering multiple domains and difficulty levels, demonstrating both robustness and generality.

5. **Limited diversity in evaluation domains:** While the proposed approach shows promising results within the PyBoy environment, its generalization to other application domains—such as robotic manipulation, as explored in [1–3]—remains untested. How well would the proposed framework transfer to such real-world or embodied settings? Expanding the evaluation beyond the current synthetic setup would substantially strengthen the work’s impact and credibility.

**Questions:**

1. Motivation and justification: Can the authors elaborate on the specific limitations of existing progress prediction or reward modeling approaches that motivate the need for this new method? What concrete evidence supports the claim that current methods are insufficient for VLM-based policy learning?

2. Comparative performance of the progress scorer: How does the proposed progress scorer quantitatively and qualitatively compare against established baselines such as Ma et al. (ICLR 2025), Hung et al. (ICLR 2025), and Rocamonde et al. (ICLR 2024)?

3. Downstream policy evaluation: Have the authors evaluated how the proposed scorer impacts downstream policy learning performance compared to re-implemented or adapted versions of existing baselines? If not, how can the authors ensure that the observed gains are indeed due to the progress scorer rather than other design differences?

4. Experimental comprehensiveness: The current experiments seem limited to a narrow task set. Could the authors expand the evaluation to include multiple domains and difficulty levels to better demonstrate the robustness and generality of the proposed framework?

5. Generality and transferability: Beyond the PyBoy environment, how well would the proposed framework generalize to real-world or embodied domains (e.g., robotic manipulation)? Are there foreseeable limitations or required adaptations to achieve such transferability?

---

> ### Author Response · Authors · 2025-11-26
>
> In response, we strengthened the motivation for RTA by clarifying why other baseline models become unstable in visually complex game-like settings and why RTA provides a more robust shaping mechanism. We expanded our experimental validation with (i) additional baselines implemented under matched data and compute, (ii) broader task coverage including multiple Catrap levels and Kirby, (iii) an explicit evaluation of reward-success alignment, (iv) downstream policy comparisons under a uniform PPO/MLP pipeline, and (v) cross-domain robustness tests using heterogeneous YouTube data.
>
> ### Motivation & justification
>
> W1 and Q1: Motivation, reasoning and evidence.
> A: In visually complex, game-like environments, we observe that common scalar reward models (e.g., Rank2Reward-style learned scalars; VLM-RM/CLIP-style goal scoring, see section Baseline comparison) can become unstable when the agent visits states that deviate from trained demonstrations which model was trained on or from the goal text (if it exists).
>
> Evidence we will surface in the introduction (and point to in experiments).
>
> - **End-to-end control** (section Baseline comparison): with matched data/compute, RTA achieves higher success than Rank2Reward and VLM-RM variants across Catrap levels and Kirby.
> - **Reward-success alignment** (section Baseline comparison): on Catrap, episode predicted reward is positively associated with success rate
> - **Generalizability on heterogeneous video sources** (section Experimental comprehensiveness & multi-domain tests): Stage-1 scorers trained on mixed YouTube playthroughs still support downstream success on held-out games/levels.
> - **Generalization across different stage 2 policies** (section Downstream policy evaluation): stage-1 ranker work good with MLP-based policy.
>
> ### Experimental comprehensiveness & multi-domain tests
>
> W4, Q4, W5, Q5: Insufficient experimental validation of the primary contribution.
>
> A: Furthermore, to test the robustness of the Stage-1 scorer under diverse video sources, we conducted an additional experiment in which RTA is trained on either a single full playthrough (Catrap and Kirby) or a collection of YouTube PyBoy playthroughs featuring varied compression games (excluding Catrap and Kirby) and then evaluate downstream on Catrap/Kirby. Despite these substantial visual differences, the scorer generalizes well across sources, which can be in table:
>
> |  | L2 | L4 | L6 | Kirby-lvl0 |
> | --- | --- | --- | --- | --- |
> | Ours (RTA) | 1.00±0.00 | 0.60±0.28 | 0.33±0.34 | 0.07±0.09 |
> | Youtube | 1.00±0.00 | 0.47±0.25 | 0.60±0.28 | 0.20±0.16 |
> | Full Catrap | 1.00±0.00 | 0.47±0.38 | 0.60±0.28 | 0.07±0.09 |
> | Full Kirby | - | - | - | 0.07±0.09 |
>
> These results indicate that the proposed progress scorer is resilient to heterogeneous video domains and maintains ordering consistency even under non-uniform visual styles.

---

> > ### Author Response · Authors · 2025-11-26
> >
> > ### Baseline comparison
> >
> > W2 and Q2: Insufficient experimental validation of the primary contribution.
> >
> > A: We implemented baselines - Rank2Reward, a VLM-RM/CLIP-style reward model (plus a regularized variant), and GVL-style (Stage-1 scorer used without our ranking training), inside the same PyBoy framework.
> >
> > To increase the breadth of our evaluation, we extended the set of PyBoy tasks by incorporating multiple Catrap levels (L2, L4, L6), deliberately including levels in which agents may enter irreversible failure states. We also added a separate game, Kirby, which presents distinct visual statistics, action dynamics, and level structures. In VideoGameBench [1] Kirby lvl-0 is a challenging benchmark on which many frontier VLMs fail (reported 0% success for several large proprietary models).
> >
> > All methods are trained on the identical expert videos and evaluated under the same observation/action spaces and rollout budgets as RTA (matched data + compute). Results are in table below.
> >
> > Success-rate comparison (mean ± std over 3 seeds):
> >
> > |  | L2 | L4 | L6 | Kirby-lvl0 |
> > | --- | --- | --- | --- | --- |
> > | Ours (RTA) | 1.00±0.00 | 0.60±0.28 | 0.33±0.34 | 0.07±0.09 |
> > | Rank2Reward [2] | 0.60±0.28 | 0.20±0.0 | 0.13±0.09 | - |
> > | VLM-RM [3] | 0.40±0.16 | 0.0±0.0 | 0.07±0.09 | 0.0±0.0 |
> > | VLM-RM (reg, \alpha=0.5) [3] | 0.27±0.19 | 0.0±0.0 | 0.0±0.0 | 0.0±0.0 |
> > | GVL [4] | 0.47±0.25 | - | - | - |
> >
> > **Regarding VICtoR:**
> >
> > VICtoR is an interesting approach, but not directly comparable to our setting: it is a vision–instruction correlation method that relies on natural-language task descriptions and additional supervision (e.g., motion-level annotations and object/state signals) and is designed for long-horizon robotic manipulation. In contrast, our setup uses only unlabeled expert video snippets in pixel-based game environments, with no instructions, object/state labels, or hierarchical annotations. We will cite and discuss VICtoR in Related Work. A faithful VICtoR adaptation would therefore change the problem definition and violate the assumptions shared by our baselines, so we do not include it in our evaluation.
> >
> > **Reward-success alignment**
> >
> > To assess whether each learned reward provides a meaningful optimization signal, we report the Pearson correlation  between mean cumulative reward and success reward the training stage. We compute this statistic for the best-performing seed of each method. When a method never solves a level, the correlation is effectively undefined; in the table this appears as 0 or “–”.
> >
> > | Method | L2 | L4 | L6 | Kirby-lvl0 |
> > | --- | --- | --- | --- | --- |
> > | RTA (ours) | 0.76 | 0.48 | 0.42 | 0.13 |
> > | Rank2Reward | −0.04 | −0.02 | −0.04 | – |
> > | VLM-RM | 0.53 | 0.00 | −0.33 | 0.00 |
> > | VLM-RM (reg., α=0.5) | 0.25 | 0.00 | 0.00 | 0.00 |
> > | GVL | −0.01 | – | – | – |
> >
> > On Catrap, RTA shows a consistently positive reward–success association (r ≈ 0.4–0.8), whereas Rank2Reward/GVL are near zero and VLM-RM is unstable across levels (positive on L2, negative on L6). On Kirby level-0 the association is weaker (r ≈ 0.13), consistent with longer horizons and more backtracking; we now highlight this in the paper as a limitation of our current setup on very long-horizon tasks. Also we attach visual examples of [good](https://8upload.com/display/5568e79f50b4075f/Frame_12.png.php) and [bad](https://8upload.com/display/37e6f5f927fc05b7/Frame_11.png.php) (in terms of success) level completion and their corresponding correlation rewards 0.77 and -0.04.

---

> > > ### Author Response · Authors · 2025-11-26
> > >
> > > ### Downstream policy evaluation
> > >
> > > W3 and Q3: Lack of baseline comparisons in policy evaluation.
> > >
> > > A:  Downstream policy evaluation (to reduce architectural confounds). To decouple the reward signal from the VLM policy class, we also run PPO with an MLP policy on Catrap (L2/L4), using the same RTA reward computation. We additionally compare two reward schedules: (i) querying every N steps vs (ii) only terminal querying. We observe that the optimal schedule differs between the VLM and MLP policies, plausibly due to different exploration dynamics: an initialized MLP benefits from termination step RTA feedback (we reward full trajectory), whereas the VLM policy often benefits from shorter-window shaping. Overall, these results are consistent with the view that RTA’s gains are driven by the reward signal’s ordering stability rather than a particular policy architecture.
> > >
> > > **Downstream PPO evaluation results** (mean ± std over 3 seeds for VLM, and over 5 seeds on MLP):
> > >
> > > |  | L2 | L4 |
> > > | --- | --- | --- |
> > > | VLM + RTA reward every N steps | 1.00 ± 0.00 | 0.60 ± 0.28 |
> > > | VLM + RTA Only-end reward | 0.33 ± 0.19 | 0.0 ± 0.0 |
> > > | MLP + RTA reward every N steps  | 0.79 ± 0.14 | 0.27± 0.21 |
> > > | MLP + RTA Only-end reward | 1.00 ± 0.00 | 1.00 ± 0.00 |
> > >
> > > These results further support the conclusion that the improvements observed with RTA primarily stem from the stability and ordering consistency of the proposed scorer, rather than from architectural or optimization differences.
> > >
> > > [1] - VideoGameBench: Can Vision-Language Models complete popular video games?
> > >
> > > [2] - Rank2Reward: Learning Shaped Reward Functions from Passive Video
> > >
> > > [3] - Vision-Language Models are Zero-Shot Reward Models for Reinforcement Learning
> > >
> > > [4] - Vision Language Models are In-Context Value Learners

---

> > > > ### Author Response · Authors · 2025-11-26
> > > >
> > > > We hope our revisions and clarifications address your concerns and would appreciate your reconsideration of the overall score. If anything remains unclear, we are ready to provide further explanation.

---

### Author Response · Authors · 2025-11-28

We thank all reviewers for their valuable comments and detailed feedback. Your suggestions enabled us to significantly improve the clarity, breadth, and rigor of the paper, without altering the core methodology or primary results. Below we outline the major enhancements made in response to your input.

**Clearer framing of the problem and contributions.**

The introduction and related work sections now provide a more explicit rationale for the rank-correlation approach, clarifying how RTA differs from prior reward-modeling and imitation-from-observation methods. The contributions are enumerated more precisely, emphasizing the generality of the correlation-only reward and the reusability of the VLM progress scorer across levels and games.

**Expanded experimental evaluation and baseline coverage.**

The experimental section has been substantially broadened to address requests for stronger baselines and wider validation. We now report direct comparisons to Rank2Reward, VLM-RM, GVL,  and supervised fine-tuning (SFT) across Catrap (levels 2, 4, and 6) and Kirby, as well as new control experiments using an MLP policy. Oracle reward comparisons and success rates are also included to strengthen the empirical grounding.

**Generalization and robustness analysis.**

We added cross-domain experiments using progress scorers trained on YouTube playthroughs from other games and players, testing RTA’s robustness to shifts in visual style and gameplay. These results demonstrate that a single scorer can generalize across tasks without per-level tuning. We conduct additional ablations on policy type, compare same scorer reward with standard MLP as policy. More results can be found in appendix.

**Improved statistical reporting and sensitivity analysis.**

To address reviewer concerns about experimental rigor, we increased the number of seeds where feasible and now report mean and standard deviation across runs for key settings. We report experiments with 3 seeds for VLM, and with 5 seeds for MLP. Additional ablations on window size, reward averaging, and reward frequency are presented to clarify the stability and reliability of the method.

**Enhanced narrative coherence and visualizations.**

Main text and figures have been reorganized for improved readability. Cross-level transfer results now appear in Section 4.2, with main control results and ablations consolidated in Sections 4.3 and 4.4. Additional tables show the alignment between progress-based reward and success rate, further validating the shaping power of our approach.

We appreciate the reviewers’ engagement and constructive feedback. The revised paper maintains all core contributions while presenting a broader and more rigorous evaluation, and we hope these improvements are reflected in your final assessments. Should any further clarification be needed during the discussion phase, we are happy to provide it.

---

### Author Response · Authors · 2025-12-04

Dear Area Chair,

Thank you for guiding the discussion. Below, we summarize the principal reviewer concerns and the targeted changes made in the revised paper. As the reviews largely focused on completeness, positioning, and empirical rigor, we believe the strengthened version now fully addresses the core concerns.

### Reviewer mxSc

Concerns:

- Lack of justification and evidence in the introduction
- Missing comparisons with prior progress scorers and policy evaluation baselines (Rank2Reward, VLM-RM, VICtoR)
- Limited experimental validation and domain diversity

What was changed:

- The introduction and related work were rewritten to clearly articulate the limitations of prior video reward methods, motivating the RTA approach and highlighting its unique aspects.
- Direct empirical comparisons were added to Rank2Reward, VLM-RM (including a regularized variant), and a GVL-style baseline (under matched data and compute).
- The experimental suite was broadened to include multiple Catrap levels (L2, L4, L6) and Kirby (a hard long-horizon task), showing that RTA outperforms all baselines across these settings.
- The experimental suite for stage 2 scorer policy model was broadened to include MLP-based policy, showing that RTA perform very good with small untrained policy model (not pretrained VLM).
- Alignment between reward and success rate was quantified, and more visualizations/tables were added to clarify this connection.
- Domain generalization was evaluated by training Stage-1 scorers on pooled or YouTube playthroughs and reusing them across levels/games.
- Statistical reporting improved: all main results now use 3 seeds or 5 seeds, reporting mean and standard deviation.

### Reviewer Pa6W

Concerns:

- The design resembles prior work; unclear how RTA performs in long-horizon or non-monotonic tasks.
- Experiments only in “simplified simulation.”
- Lack of comparison with recent offline RL or IRL methods using expert demos (Rank2Reward).

What was changed:

- The scope of experiments was expanded to harder Catrap levels and the long-horizon Kirby task, directly addressing robustness to non-monotonic and backtracking-heavy settings.
- Rank2Reward, VLM-RM (in 2 variations), and GVL baselines were implemented and evaluated in the same regime as RTA, showing that only RTA remains stable and successful off-demonstration.
- The method section, experiments, and related work now explicitly position RTA relative to both video reward models and imitation-from-observation methods, explaining why certain baselines (VICtoR) are not directly comparable due to additional supervision or action labels.
- Cross-domain generalization was tested by using heterogeneous video sources for scorer training.

### Reviewer jdVY

Concerns:

- Insufficient statistical rigor (too few seeds, no error bars/confidence intervals)
- Evaluation limited to Catrap; broader policy experiments needed (Atari/robotics suggested)
- Missing ablations: window size, query frequency, anchor/no-anchor, shuffles, etc.
- Reward-success alignment and the role of non-Markovian rewards not adequately analyzed

What was changed:

- All main policy experiments are now reported with 3 seeds or 5 seeds, including means and standard deviations.
- Policy evaluation was extended to multiple Catrap levels and Kirby; a non-VLM MLP policy was also evaluated using RTA reward.
- Direct empirical comparisons were added to Rank2Reward, VLM-RM (including a regularized variant), and a GVL-style baseline (under matched data and compute).
- Comprehensive ablation studies were added: window length, reward frequency, number of shuffles, with tabular results.
- Reward-success alignment was measured (episode-wise Spearman correlation); a comparison to an oracle sparse reward was included.
- The PPO/MDP discussion was clarified, explaining why the windowed reward does not introduce new non-Markovian artifacts compared to existing LLM/VLM agent work.

---

> ### Author Response · Authors · 2025-12-04
>
> ### Reviewer UiJZ
>
> Concerns:
>
> - Would like to see success rate/oracle reward comparisons in learning curves/tables
> - Request for a non-VLM ranking baseline (Rank2Reward) and a non-VLM policy class (MLP)
> - More detailed SFT baseline description, and discussion of policy class effects
>
> What was changed:
>
> - All main experiments now include success rate curves and oracle reward baselines.
> - Rank2Reward (CNN-based), VLM-RM (CLIP-based), and a regularized variant are now compared directly to RTA.
> - An MLP policy baseline was added, showing that RTA’s gains are due to the reward, not the policy architecture.
> - More details were added for the SFT baseline and evaluation methodology.
> - Cross-domain generalization was tested by using heterogeneous video sources for scorer training.
>
> General improvements:
>
> - Most sections were rewritten for greater clarity and precision (introduction, results, and related work).
> - Visual examples and quantitative tables were added or enhanced to directly address reviewer requests.
> - Cross-domain generalization and robustness analysis were substantially expanded.
> - The ablation appendix was added and clearly referenced.
>
> We believe the revised submission is much stronger, with broader coverage and sharper positioning, and addresses all major reviewer concerns. We respectfully ask the committee to consider these improvements in your final decision.

---

### Meta-Review · Area_Chair_BwuM · 2026-01-07

**Summary:**

There was consensus among reviewers regarding the lack of empirical and statistical rigor required for a clear accept. RTA proposes a two-stage method: Stage 1 trains a VLM as a progress scorer on shuffled expert video frames, and Stage 2 uses spearman rank correlation between predicted progress and timestamps as a reward for online policy learning.

Reviewers expressed critical concerns about the statistical significance of the results, as the original submission relied on only 2 seeds (during rebuttal they re-ran with 3 seeds despite asking for 5 seeds). Furthermore, the evaluation was initially limited to a single game environment (Catrap), leading to questions about the method's ability to generalize to complex/diverse tasks. While the authors provided a high effort rebuttal - including new baselines and harder levels - the narrow environmental diversity and statistical rigor makes the experimental validation limited. This paper is exploring an interesting line of work but could be stronger after another round of iteration. It is interesting and important to investigate a broader set of environment and ask - where does this method fail and where are the underlying assumptions violated?

**Reviewer Concerns:**

Concerns addressed:
- The authors added direct comparisons against Rank2Reward, VLM-RM, and GVL-style baselines, showing that RTA outperforms them in the tested environments.
- The experimental suite was expanded beyond basic Catrap levels to include levels with dead-ends (L4, L6) and the long-horizon game Kirby.
- To decouple the reward signal from the actor architecture, the authors included successful experiments using an MLP-based policy.
- Evidence was provided that a scorer trained on pooled or Youtube scraped video can support zero-shot transfer to new levels/games.

Outstanding concerns:
- jdVY explicitly required 5 seeds for publication. The authors only provided 3 seeds for the primary VLM policy. This may be a relatively minor issue but coupled with limited environment diversity, it is difficult to understand the generalizability of this approach.
- The method remains unvalidated outside of 2D discrete action games. Its performance in high-dimensional, continuous control, or highly stochastic settings is still unproven.
- The reward depends on a sliding window of history that the policy does not fully observe, creating a theoretical mismatch with the PPO algorithm that was not fully resolved.

**Reviewer Scores:**

- mxSc: This reviewer appreciated the competitive baselines but remained concerned about evaluation diversity.

- Pa6W: This reviewer skeptical that local frame order consistency is a sufficient proxy for global progress in complex tasks.

- jdVY: While acknowledging the improved clarity, this reviewer wanted to see more statistical rigor.

- UiJZ: Valued the added MLP and oracle reward comparisons but shared concerns regarding task-specific supervision assumptions.

---

### Decision · Program_Chairs · 2026-01-26

Reject